# Differentiable Top-$k$ with Optimal Transport

**Yujia Xie**[*]
College of Computing
Georgia Tech
Xie.Yujia000@gmail.com

**Hanjun Dai**
Google Brain
hadai@google.com

**Minshuo Chen**
College of Engineering
Georgia Tech
mchen393@gatech.edu

**Bo Dai**
Google Brain
bodai@google.com

**Tuo Zhao**
College of Engineering
Georgia Tech
tourzhao@gatech.edu

**Hongyuan Zha**[†]
School of Data Science
Shenzhen Research Institute
of Big Data, CUHK, Shenzhen
zhahy@cuhk.edu.cn

**Wei Wei**
Google Cloud AI
wewei@google.com

**Tomas Pfister**
Google Cloud AI
tpfister@google.com

## Abstract

The top-$k$ operation, i.e., finding the $k$ largest or smallest elements from a collection of scores, is an important model component, which is widely used in information retrieval, machine learning, and data mining. However, if the top-$k$ operation is implemented in an algorithmic way, e.g., using bubble algorithm, the resulting model cannot be trained in an end-to-end way using prevalent gradient descent algorithms. This is because these implementations typically involve swapping indices, whose gradient cannot be computed. Moreover, the corresponding mapping from the input scores to the indicator vector of whether this element belongs to the top-$k$ set is essentially discontinuous. To address the issue, we propose a smoothed approximation, namely the SOFT (Scalable Optimal transport-based diFferenTiable) top-$k$ operator. Specifically, our SOFT top-$k$ operator approximates the output of the top-$k$ operation as the solution of an Entropic Optimal Transport (EOT) problem. The gradient of the SOFT operator can then be efficiently approximated based on the optimality conditions of EOT problem. We apply the proposed operator to the $k$-nearest neighbors and beam search algorithms, and demonstrate improved performance.

## 1 Introduction

The top-$k$ operation, i.e., finding the $k$ largest or smallest elements from a set, is widely used for predictive modeling in information retrieval, machine learning, and data mining. For example, in image retrieval (Babenko et al., 2014; Radenović et al., 2016; Gordo et al., 2016), one needs to query the $k$ nearest neighbors of an input image under certain metrics; in the beam search (Reddy et al., 1977; Wiseman and Rush, 2016) algorithm for neural machine translation, one needs to find the $k$ sequences of largest likelihoods in each decoding step.

---

[*]Work done in a Google internship.

[†]Also affliated with Shenzhen Institute of Artificial Intelligence and Robotics for Society. On leave from College of Computing, Georgia Tech.

Although the ubiquity of top-$k$ operation continues to grow, the operation itself is difficult to be integrated into the training procedure of a predictive model. For example, we consider a neural network-based $k$-nearest neighbor classifier. Given an input, we use the neural network to extract features from the input. Next, the extracted features are fed into the top-$k$ operation for identifying the $k$ nearest neighbors under some distance metric. We then obtain a prediction based on the $k$ nearest neighbors of the input. In order to train such a model, we choose a proper loss function, and minimize the average loss across training samples using (stochastic) first-order methods. This naturally requires the loss function being differentiable with respect to the input at each update step. Nonetheless, the top-$k$ operation does not exhibit an explicit mathematical formulation: most implementations of the top-$k$ operation, e.g., bubble algorithm and QUICKSELECT (Hoare, 1961), involve operations on indices such as indices swapping. Consequently, the training objective is difficult to formulate explicitly.

Alternative perspective — taking the top-$k$ operation as an operator — still cannot resolve the differentibility issue. Specifically, the top-$k$ operator[3] maps a set of inputs $x_1, \ldots, x_n$ to an index vector $\{0,1\}^n$. Whereas the Jacobian matrix of such a mapping is not well defined. As a simple example, consider two scalars $x_1, x_2$. The top-1 operation as in Figure 1 returns a vector $[A_1, A_2]^\top$, with each entry denoting whether the scalar is the larger one (1 for true, 0 for false). Denote $A_1 = f(x_1, x_2)$. For a fixed $x_2$, $A_1$ jumps from 0 to 1 at $x_1 = x_2$. It is clear that $f$ is not differentiable at $x_1 = x_2$, and the derivative is identically zero otherwise.

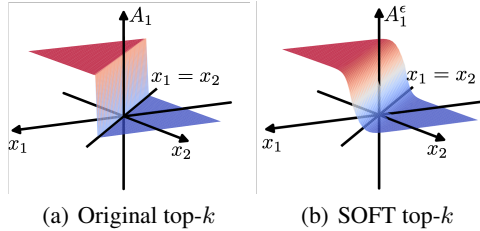

(a) Original top-$k$    (b) SOFT top-$k$

**Figure 1:** Illustration of the top-$k$ operators.

Due to the aforementioned difficulty, existing works resort to two-stage training for models with the top-$k$ operation. We consider the neural network-based $k$-nearest neighbor classifier again. As proposed in Papernot and McDaniel (2018), one first trains the neural network using some surrogate loss on the extracted features, e.g., using softmax activation in the output layer and the cross-entropy loss. Next, one uses the $k$-nearest neighbor for prediction based on the features extracted by the well-trained neural network. This training procedure, although circumventing the top-$k$ operation, makes the training and prediction misaligned; and the actual performance suffers.

In this work, we propose the SOFT (Scalable Optimal transport-based diFferenTiable) top-$k$ operation as a differentiable approximation of the standard top-$k$ operation in Section 2. Specifically, motivated by the implicit differentiation (Duchi et al., 2008; Griewank and Walther, 2008; Amos and Kolter, 2017; Luise et al., 2018) techniques, we first parameterize the top-$k$ operation in terms of the optimal solution of an Optimal Transport (OT) problem. Such a re-parameterization is still not differentiable with respect to the input. To rule out the discontinuity, we impose entropy regularization to the optimal transport problem, and show that the optimal solution to the Entropic OT (EOT) problem yields a differentiable approximation to the top-$k$ operation. Moreover, we prove that under mild assumptions, the approximation error can be properly controlled.

We then develop an efficient implementation of the SOFT top-$k$ operation in Section 3. Specifically, we solve the EOT problem via the Sinkhorn algorithm (Cuturi, 2013). Given the optimal solution, we can explicitly formulate the gradient of SOFT top-$k$ operation using the KKT (Karush-Kuhn-Tucker) condition. As a result, the gradient at each update step can be efficiently computed with complexity $\mathcal{O}(n)$, where $n$ is the number of elements in the input set to the top-$k$ operation.

Our proposed SOFT top-$k$ operation allows end-to-end training, and we apply SOFT top-$k$ operation to $k$NN for classification in Section 4 and beam search in Section 5. The experimental results demonstrate significant performance gain over competing methods, as an end-to-end training procedure resolves the misalignment between training and prediction.

**Notations.** We denote $\|\cdot\|_2$ as the $\ell_2$ norm of vectors, $\|\cdot\|_F$ as the Frobenius norm of matrices. Given two matrices $B, D \in \mathbb{R}^{n \times m}$, we denote $\langle B, D \rangle$ as the inner product, i.e., $\langle B, D \rangle = \sum_{i=1,j=1}^{n,m} B_{ij} D_{ij}$. We denote $B \odot D$ as the element-wise multiplication of $B$ and $D$. We denote $\mathbb{1}(\cdot)$ as the indicator function, i.e., the output of $\mathbb{1}(\cdot)$ is 1 if the input condition is satisfied, and is 0 otherwise. For matrix $B \in \mathbb{R}^{n \times m}$, we denote $B_{i,:}$ as the $i$-th row of the matrix.

The softmax function for matrix $B$ is defined as $\mathrm{softmax}_i(B_{ij}) = e^{B_{ij}}/\sum_{\ell=1}^{n} e^{B_{lj}}$. For a vector $b \in \mathbb{R}^n$, we denote $\mathrm{diag}(b)$ as the matrix where the $i$-th diagonal entries is $b_i$.

## 2  SOFT Top-$k$ Operator

We adopt the following definition of the (augment of) top-$k$ operator. Given a set of scalars $\mathcal{X} = \{x_i\}_{i=1}^{n}$, the standard top-$k$ operator returns a vector $A = [A_1, \dots, A_n]^\top$, such that

$$A_i = \begin{cases} 1, & \text{if } x_i \text{ is a top-}k \text{ element in } \mathcal{X}, \\ 0, & \text{otherwise.} \end{cases}$$

Note that the definition is essentially an "arg-top-$k$" operation since it marks the top-$k$ *indices* as 1, instead of returning the top-$k$ *values*. This allows more flexibility since we can obtain the top-$k$ values by multiplying $A$ to $\mathcal{X}$. The goal is to design a smooth relaxation of the standard top-$k$ operator. Without loss of generality, we refer to top-$k$ elements as the *smallest $k$ elements*.

### 2.1  Parameterizing Top-$k$ Operator as OT Problem

We first show that the standard top-$k$ operator can be parameterized in terms of the solution of an Optimal Transport (OT) problem (Monge, 1781; Kantorovich, 1960). We briefly introduce OT problems for self-containedness. An OT problem finds a transport plan between two distributions, while the expected cost of the transportation is minimized. We consider two discrete distributions defined on supports $\mathcal{A} = \{a_i\}_{i=1}^{n}$ and $\mathcal{B} = \{b_j\}_{j=1}^{m}$, respectively. Denote $\mathbb{P}(\{a_i\}) = \mu_i$ and $\mathbb{P}(\{b_j\}) = \nu_j$, and let $\mu = [\mu_1, \dots, \mu_n]^\top$ and $\nu = [\nu_1, \dots, \nu_m]^\top$. We further denote $C \in \mathbb{R}^{n \times m}$ as the cost matrix with $C_{ij}$ being the cost of transporting mass from $a_i$ to $b_j$. An OT problem can be formulated as

$$\Gamma^* = \underset{\Gamma \geq 0}{\mathrm{argmin}} \langle C, \Gamma \rangle, \quad \text{s.t., } \Gamma \mathbf{1}_m = \mu, \; \Gamma^\top \mathbf{1}_n = \nu, \tag{1}$$

where $\mathbf{1}$ denotes a vector of ones. The optimal $\Gamma^*$ is referred to as the *optimal transport plan*.

In order to parameterize the top-$k$ operator using the optimal transport plan $\Gamma^*$, we set the support $\mathcal{A} = \mathcal{X}$ and $\mathcal{B} = \{0, 1\}$ in (1), with $\mu, \nu$ defined as

$$\mu = \mathbf{1}_n/n, \quad \nu = [k/n, (n-k)/n]^\top.$$

We take the cost to be the squared Euclidean distance, i.e., $C_{i1} = x_i^2$ and $C_{i2} = (x_i - 1)^2$ for $i = 1, \dots, n$. We then establish the relationship between the output $A$ of the top-$k$ operator and $\Gamma^*$.

**Proposition 1.** Consider the setup in the previous paragraph. Without loss of generality, we assume $\mathcal{X}$ has no duplicates. Then the optimal transport plan $\Gamma^*$ of (1) is

$$\Gamma^*_{\sigma_i, 1} = \begin{cases} 1/n, & \text{if } i \leq k, \\ 0, & \text{if } k+1 \leq i \leq n. \end{cases}, \quad \Gamma^*_{\sigma_i, 2} = \begin{cases} 0, & \text{if } i \leq k, \\ 1/n, & \text{if } k+1 \leq i \leq n, \end{cases} \tag{2}$$

with $\sigma$ being the sorting permutation, i.e., $x_{\sigma_1} < x_{\sigma_2} < \cdots < x_{\sigma_n}$. Moreover, we have

$$A = n\Gamma^* \cdot [1, 0]^\top. \tag{3}$$

The proof can be found in Appendix A. Figure 3(a) illustrates the corresponding optimal transport plan for parameterizing the top-5 operator applied to a set of 7 elements. As can be seen, the mass from the 5 closest points is transported to 0, and meanwhile the mass from the 2 remaining points is transported to 1. Therefore, the optimal transport plan exactly indicates the top-5 elements.

### 2.2  Smoothing by Entropy Regularization

We next rule out the discontinuity of (1) to obtain a smoothed approximation to the top-$k$ operator.

Specifically, we employ entropy regularization to the OT problem (1):

$$\Gamma^{*,\epsilon} = \underset{\Gamma \geq 0}{\mathrm{argmin}} \langle C, \Gamma \rangle + \epsilon H(\Gamma), \quad \text{s.t., } \Gamma \mathbf{1}_m = \mu, \; \Gamma^\top \mathbf{1}_n = \nu, \tag{4}$$

where $h(\Gamma) = \sum_{i,j} \Gamma_{ij} \log \Gamma_{ij}$ is the entropy regularizer. We define $A^\epsilon = n\Gamma^{*,\epsilon} \cdot [0, 1]^\top$ as a smoothed counterpart of output $A$ in the standard top-$k$ operator. Accordingly, SOFT top-$k$ operator is defined as the mapping from $\mathcal{X}$ to $A^\epsilon$. We show that the Jacobian matrix of SOFT top-$k$ operator exists and is nonzero in the following theorem.

**Theorem 1.** For any $\epsilon > 0$, SOFT top-$k$ operator: $\mathcal{X} \mapsto A^\epsilon$ is differentiable, as long as the cost $C_{ij}$ is differentiable with respect to $x_i$ for any $i, j$. Moreover, the Jacobian matrix of SOFT top-$k$ operator always has a nonzero entry for any $\mathcal{X} \in \mathbb{R}^n$.

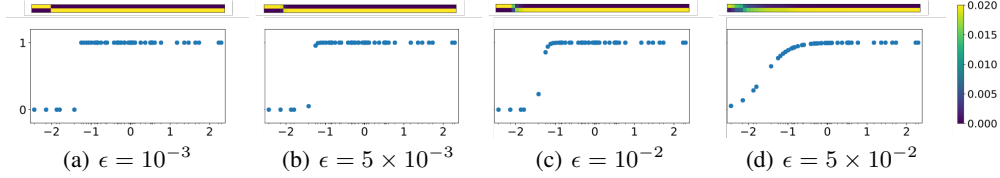

(a) $\epsilon = 10^{-3}$     (b) $\epsilon = 5 \times 10^{-3}$     (c) $\epsilon = 10^{-2}$     (d) $\epsilon = 5 \times 10^{-2}$

**Figure 2:** Color maps of $\Gamma^\epsilon$ (upper) and the corresponding scatter plots of values in $A^\epsilon$ (lower), where $\mathcal{X}$ contains 50 standard Gaussian samples, and $K = 5$. The scatter plots show the correspondence of the input $\mathcal{X}$ and output $A^\epsilon$.

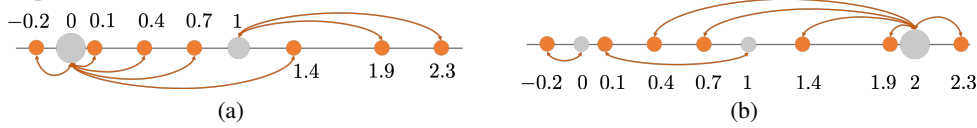

(a)                                            (b)

**Figure 3:** (a). Illustration of the OT plan with input $\mathcal{X} = [0.4, 0.7, 2.3, 1.9, -0.2, 1.4, 0.1]^\top$ and $k = 5$. We set $\nu = [\frac{5}{7}, \frac{2}{7}]^\top$. In this way, 5 of the 7 scores align with 0, while $\{2.3, 1.9\}$ align with 1. (b). Illustration for *sorted* top-$k$ with similar input and $k = 2$. We set $\nu = [\frac{1}{7}, \frac{1}{7}, \frac{5}{7}]^\top$ and $\mathcal{B} = [0, 1, 2]^\top$. Then, the smallest score $-0.2$ aligns with 0, the second smallest score 0.1 aligns with 1, and the rest of the scores align with 2.

The proof can be found in Appendix A. We remark that the entropic OT (4) is computationally more friendly, since it allows the usage of first-order algorithms (Cuturi, 2013).

The Entropic OT introduces bias to the SOFT top-$k$ operator. The following theorem shows that such a bias can be effectively controlled.

**Theorem 2.** Given a distinct sequence $\mathcal{X}$ and its sorting permutation $\sigma$, with Euclidean square cost function, for the proposed top-$k$ solver we have

$$\|\Gamma^{*,\epsilon} - \Gamma^*\|_{\mathrm{F}} \leq \frac{\epsilon(\ln n + \ln 2)}{n(x_{\sigma_{k+1}} - x_{\sigma_k})}.$$

Therefore, with a small enough $\epsilon$, the output vector $A^\epsilon$ can well approximate $A$, especially when there is a large gap between $x_{\sigma_k}$ and $x_{\sigma_{k+1}}$. Besides, Theorem 2 suggests a trade-off between the bias and regularization of SOFT top-$k$ operator. See Section 7 for a detailed discussion.

### 2.3 Sorted SOFT Top-$k$ Operator

In some applications, we not only need to distinguish the top-$k$ elements, but also sort the top-$k$ elements. For example, in image retrieval (Gordo et al., 2016), the retrieved $k$ images are expected to be sorted. Our SOFT top-$k$ operator can be extended to the sorted SOFT top-$k$ operator.

Analogous to the derivation of the SOFT top-$k$ operator, we first parameterize the sorted top-$k$ operator in terms of an OT problem. Specifically, we keep $\mathcal{A} = \mathcal{X}$ and $\mu = \mathbf{1}_n/n$ and set

$$\mathcal{B} = [0, 1, 2, \cdots, k]^\top, \text{ and } \nu = [1/n, \cdots, 1/n, (n-k)/n]^\top.$$

One can check that the optimal transport plan of the above OT problem transports the smallest element in $\mathcal{A}$ to 0 in $\mathcal{B}$, the second smallest element to 1, and so on so forth. This in turn yields the sorted top-$k$ elements. Figure 3(b) illustrates the sorted top-2 operator and its optimal transport plan.

The sorted SOFT top-$k$ operator is obtained similarly to SOFT top-$k$ operator by solving the entropy regularized OT problem. We can show that the sorted SOFT top-$k$ operator is differentiable and the bias can be properly controlled.

## 3 Efficient Implementation

We now present our implementation of SOFT top-$k$ operator, which consists of 1) computing $A^\epsilon$ from $\mathcal{X}$ and 2) computing the Jacobian matrix of $A^\epsilon$ with respect to $\mathcal{X}$. We refer to 1) as the forward pass and 2) as the backward pass.

**Forward Pass** The forward pass from $\mathcal{X}$ to $A^\epsilon$ can be efficiently computed using Sinkhorn algorithm.

---

**Algorithm 1** SOFT Top-$k$

**Require:** $\mathcal{X} = [x_i]_{i=1}^n, k, \epsilon, L$
    $\mathcal{Y} = [y_1, y_2]^\top = [0, 1]^\top$
    $\mu = \mathbf{1}_n/n, \nu = [k/n, (n-K)/n]^\top$
    $C_{ij} = |x_i - y_j|^2, G_{ij} = e^{-\frac{C_{ij}}{\epsilon}}, q = \mathbf{1}_2/2$
    **for** $l = 1, \cdots, L$ **do**
       $p = \mu/(Gq), q = \nu/(G^\top p)$
    **end for**
    $\Gamma = \text{diag}(p) \odot G \odot \text{diag}(q)$
    $A^\epsilon = n\Gamma \cdot [0, 1]^\top$

---

Specifically, we run iterative Bregman projections (Benamou et al., 2015), where at the $\ell$-th iteration, we update

$$p^{(\ell+1)} = \frac{\mu}{Gq^{(\ell)}}, \quad q^{(\ell+1)} = \frac{\nu}{G^\top p^{(\ell+1)}}.$$

Here, the division is entrywise, $q^{(0)} = \mathbf{1}_2/2$, and $G \in \mathbb{R}^{n \times m}$ with $G_{ij} = e^{-\frac{C_{ij}}{\epsilon}}$. Denote $p^*$ and $q^*$ as the stationary point of the Bregman projections. The optimal transport plan $\Gamma^{*,\epsilon}$ can be obtained by $\Gamma^{*,\epsilon}_{ij} = p_i^* G_{ij} q_j^*$. The algorithm is summarized in Algorithm 1.

**Backward Pass.** Given $A^\epsilon$, we compute the Jacobian matrix $\frac{dA^\epsilon}{d\mathcal{X}}$ using implicit differentiation and differentiable programming techinques. Specifically, the Lagrangian function of Problem (4) is

$$\mathcal{L} = \langle C, \Gamma \rangle - \xi^\top (\Gamma \mathbf{1}_m - \mu) - \zeta^\top (\Gamma^\top \mathbf{1}_n - \nu) + \epsilon H(\Gamma),$$

where $\xi$ and $\zeta$ are dual variables. The KKT condition implies that $\Gamma^{*,\epsilon}$ can be formulated using the optimal dual variables $\xi^*$ and $\zeta^*$ as (Sinkhorn's scaling theorem, Sinkhorn and Knopp (1967)),

$$\Gamma^{*,\epsilon} = \mathrm{diag}(e^{\frac{\xi^*}{\epsilon}}) e^{-\frac{C}{\epsilon}} \mathrm{diag}(e^{\frac{\zeta^*}{\epsilon}}). \tag{5}$$

Substituting (5) into the Lagrangian function, we obtain

$$\mathcal{L}(\xi^*, \zeta^*; C) = (\xi^*)^\top \mu + (\zeta^*)^\top \nu - \epsilon \sum_{i,j=1}^{n,m} e^{-\frac{C_{ij} - \xi_i^* - \zeta_j^*}{\epsilon}}.$$

We now compute the gradient of $\xi^*$ and $\zeta^*$ with respect to $C$, such that we can obtain $d\Gamma^{*,\epsilon}/dC$ by the chain rule applied to (5). Denote $\omega^* = [(\xi^*)^\top, (\zeta^*)^\top]^\top$, and $\phi(\omega^*; C) = \partial \mathcal{L}(\omega^*; C)/\partial \omega^*$. At the optimal dual variable $\omega^*$, the KKT condition immediately yields

$$\phi(\omega^*; C) \equiv 0.$$

By the chain rule, we have

$$\frac{d\phi(\omega^*; C)}{dC} = \frac{\partial \phi(\omega^*; C)}{\partial C} + \frac{\partial \phi(\omega^*; C)}{\partial \omega^*} \frac{d\omega^*}{dC} = 0.$$

Rearranging terms, we obtain

$$\frac{d\omega^*}{dC} = -\left(\frac{\partial \phi(\omega^*; C)}{\partial \omega^*}\right)^{-1} \frac{\partial \phi(\omega^*; C)}{\partial C}. \tag{6}$$

Combining (5), (6), $C_{ij} = (x_i - y_j)^2$, and $A^\epsilon = n\Gamma^{*,\epsilon} \cdot [1,0]^\top$, the Jacobian matrix $dA^\epsilon/d\mathcal{X}$ can then be derived using the chain rule again.

The detailed derivation and the corresponding algorithm for computing the Jacobian matrix can be found in Appendix B. The time and space complexity of the derived algorithm is $\mathcal{O}(n)$ and $\mathcal{O}(kn)$ for top-$k$ and sorted top-$k$ operators, respectively. We also include a Pytorch Paszke et al. (2017) implementation of the forward and backward pass in Appendix B by extending the `autograd` automatic differentiation package.

# 4   $k$-NN for Image Classification

The proposed SOFT top-$k$ operator enables us to train an end-to-end neural network-based $k$NN classifier. Specifically, we receive training samples $\{Z_i, y_i\}_{i=1}^N$ with $Z_i$ being the input data and $y_i \in \{1, \ldots, M\}$ the label from $M$ classes. During the training, for an input data $Z_j$ (also known as the query sample), we associate a loss as follows. Denote $Z_{\setminus j}$ as all the input data excluding $Z_j$ (also known as the template samples). We use a neural network $f_\theta$ parameterized by $\theta$ to extract features from all the input data, and measure the pairwise Euclidean distances between the extracted features of $Z_{\setminus j}$ and that of $Z_j$. Denote $\mathcal{X}_{\setminus j, \theta}$ as the collection of these pairwise distances, i.e.,

$$\mathcal{X}_{\setminus j, \theta} = \{\|f_\theta(Z_1) - f_\theta(Z_j)\|_2, \ldots, \|f_\theta(Z_{j-1}) - f_\theta(Z_j)\|_2,$$
$$\|f_\theta(Z_{j+1}) - f_\theta(Z_j)\|_2, \ldots, \|f_\theta(Z_N) - f_\theta(Z_j)\|_2\},$$

where the subscript of $\mathcal{X}$ emphasizes its dependence on $\theta$.

Next, we apply SOFT top-$k$ operator to $\mathcal{X}_{\setminus j, \omega}$, and the returned vector is denoted by $A_{\setminus j, \theta}^\epsilon$. Let $Y_{\setminus j} \in \mathbb{R}^{M \times (N-1)}$ be the matrix by concatenating the one-hot encoding of labels $y_i$ for $i \neq j$ as columns, and $Y_j \in \mathbb{R}^M$ the one-hot encoding of the label $y_j$. The loss of $Z_j$ is defined as

$$\ell(Z_j, y_j) = Y_j^\top Y_{\setminus j}^\top A_{\setminus j, \theta}^\epsilon.$$

Consequently, the training loss is $\mathcal{L}(\{Z_j, y_j\}_{j=1}^N) = \frac{1}{N}\sum_{j=1}^N \ell(Z_j, y_j)$. Recall that the Jacobian matrix of $A_{\backslash j,\theta}^\epsilon$ exists and has no zero entries. This allows us to utilize stochastic gradient descent algorithms to update $\theta$ in the neural network. Moreover, since $N$ is often large, to ease the computation, we randomly sample a batch of samples to compute the stochastic gradient at each iteration.

In the prediction stage, we use all the training samples to obtain a predicted label of a query sample. Specifically, we feed the query sample into the neural network to extract its features, and compute pairwise Euclidean distances to all the training samples. We then run the standard $k$NN algorithm (Hastie et al., 2009) to obtain the predicted label.

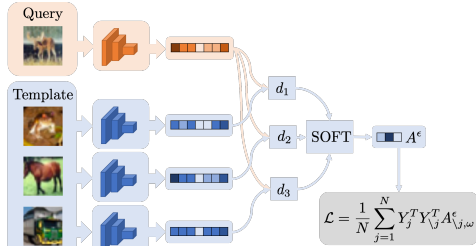

**Figure 4:** Illustration of the entire forward pass of $k$NN.

**Table 1:** Classification accuracy of kNN.

| Algorithm | MNIST | CIFAR10 |
|---|---|---|
| $k$NN | 97.2% | 35.4% |
| $k$NN+PCA | 97.6% | 40.9% |
| $k$NN+AE | 97.6% | 44.2% |
| $k$NN+pretrained CNN | 98.4% | 91.1% |
| RelaxSubSample | 99.3% | 90.1% |
| $k$NN+NeuralSort | **99.5**% | 90.7% |
| $k$NN+Cuturi et al. (2019) | 99.0% | 84.8% |
| $k$NN+Softmax $k$ times | 99.3% | 92.2% |
| CE+CNN (He et al., 2016) | 99.0% | 91.3% |
| $k$NN+**SOFT Top-**$k$ | 99.4% | **92.6**% |

## 4.1 Experiment

We evaluate the performance of the proposed neural network-based $k$NN classifier on two benchmark datasets: MNIST dataset of handwritten digits (LeCun et al., 1998) and the CIFAR-10 dataset of natural images (Krizhevsky et al., 2009) with the canonical splits for training and testing without data augmentation. We adopt the coefficient of entropy regularizer $\epsilon = 10^{-3}$ for MNIST dataset and $\epsilon = 10^{-5}$ for CIFAR-10 dataset. Further implementation details can be found in Appendix C.

**Baselines.** We consider several baselines:

1. Standard $k$NN method.
2. Two-stage training methods: we first extract the features of the images, and then perform $k$NN on the features. The feature is extracted using Principle Component Analysis (PCA, top-50 principle components is adopted), autoencoder (AE), or a pretrained Convolutional Neural Network (CNN) using the Cross-Entropy (CE) loss.
3. Differentiable *ranking* + $k$NN: This includes NeuralSort (Grover et al., 2019) and Cuturi et al. (2019). Cuturi et al. (2019) is not directly applicable, which requires adaptations (see Appendix C).
4. Stochastic $k$NN with Gumbel top-$k$ relaxation (Xie and Ermon, 2019): The model is referred as RelaxSubSample.
5. Softmax Augmentation for smoothed top-$k$ operation: A combination of $k$ softmax operation is used to replace the top-$k$ operator. Specifically, we recursively perform softmax on $\mathcal{X}$ for $k$ times (Similar idea appears in Plötz and Roth (2018)). At the $k$-th iteration, we mask the top-$(k-1)$ entries with negative infinity.
6. CNNs trained with CE without any top-$k$ component[4].

For the pretrained CNN and CNN trained with CE, we adopt identical neural networks as our method.
**Results.** We report the classification accuracies on the standard test sets in Table 1. On both datasets, the SOFT $k$NN classifier achieves comparable or better accuracies.

## 5 Beam Search for Machine Translation

Beam search is a popular method for the *inference* of Neural Language Generation (NLG) models, e.g., machine translation models. Here, we propose to incorporate beam search into the *training* procedure based on SOFT top-$k$ operator.

## 5.1 Misalignment between Training and Inference

Denote the predicted sequence as $y = [y^{(1)}, \cdots, y^{(T)}]$, and the vocabularies as $\{z_1, \cdots, z_V\}$. Consider a recurrent network based NLG model. The output of the model at the $t$-th decoding step is a probability simplex $[\mathbb{P}(y^{(t)} = z_i | h^{(t)})]_{i=1}^V$, where $h^{(t)}$ is the hidden state associated with the sequence $y^{(1:t)} = [y^{(1)}, ..., y^{(t)}]$.

Beam search recursively keeps the sequences with the $k$ largest likelihoods, and discards the rest. Specifically, at the $(t+1)$-th decoding step, we have $k$ sequences $\widetilde{y}^{(1:t),i}$'s obtained at the $t$-th step, where $i = 1, ..., k$ indexes the sequences. The likelihood of $\widetilde{y}^{(1:t),i}$ is denoted by $\mathcal{L}_{\mathrm{s}}(\widetilde{y}^{(1:t),i})$. We then select the next $k$ sequences by varying $i = 1, \ldots, k$ and $j = 1, \ldots, V$:

$$\{\widetilde{y}^{(1:t+1),\ell}\}_{\ell=1}^k = \arg\text{top-k}_{[\widetilde{y}^{(1:t),i}, z_j]} \mathcal{L}_{\mathrm{s}}([\widetilde{y}^{(1:t),i}, z_j]).$$

where $\mathcal{L}_{\mathrm{s}}([\widetilde{y}^{(1:t),i}, z_j])$ is the likelihood of the sequence appending $z_j$ to $\widetilde{y}^{(1:t),i}$ defined as

$$\mathcal{L}_{\mathrm{s}}([\widetilde{y}^{(1:t),i}, z_j]) = \mathbb{P}(y^{(t+1)} = z_j | h^{(t+1),i}) \mathcal{L}_{\mathrm{s}}(\widetilde{y}^{(1:t),i}), \tag{7}$$

and $h^{(t+1),i}$ is the hidden state generated from $\widetilde{y}^{(1:t),i}$. Note that $z_j$'s and $\widetilde{y}^{(1:t),i}$'s together yield $Vk$ choices. Here we abuse the notation: $\widetilde{y}^{(1:t+1),\ell}$ denotes the $\ell$-th selected sequence at the $(t+1)$-th decoding step, and is not necessarily related to $\widetilde{y}^{(1:t),i}$ at the $t$-th decoding step, even if $i = \ell$.

For $t = 1$, we set $\widetilde{y}^{(1)} = z_{\mathrm{s}}$ as the start token, $\mathcal{L}_{\mathrm{s}}(y^{(1)}) = 1$, and $h^{(1)} = h_{\mathrm{e}}$ as the output of the encoder. We repeat the above procedure, until the end token is selected or the pre-specified max length is reached. At last, we select the sequence $y^{(1:T),*}$ with the largest likelihood as the prediction.

Moreover, the most popular training procedure for NLG models directly uses the so-called "*teacher forcing*" framework. As the ground truth of the target sequence (i.e., gold sequence) $\bar{y} = [\bar{y}^{(1)}, \cdots, \bar{y}^{(T)}]$ is provided at the training stage, we can directly maximize the likelihood

$$\mathcal{L}_{\mathrm{tf}} = \prod_{t=1}^T \mathbb{P}(y^{(t)} = \bar{y}^{(t)} | h^{(t)}(\bar{y}^{(1:t-1)})). \tag{8}$$

As can be seen, such a training framework only involve the gold sequence, and cannot take the uncertainty of the recursive exploration of the beam search into consideration. Therefore, it yields a misalignment between model training and inference (Bengio et al., 2015), which is also referred as *exposure bias* (Wiseman and Rush, 2016).

## 5.2 Differential Beam Search with Sorted SOFT Top-$k$

To mitigate the aforementioned misalignment, we propose to integrate beam search into the training procedure, where the top-$k$ operator in the beam search algorithm is replaced with our proposed sorted SOFT top-$k$ operator proposed in Section 2.3.

Specifically, at the $(t+1)$-th decoding step, we have $k$ sequences denoted by $E^{(1:t),i}$, where $i = 1, ..., k$ indexes the sequences. Here $E^{(1:t),i}$ consists of a sequence of $D$-dimensional vectors, where $D$ is the embedding dimension. We are not using the tokens, and the reason behind will be explained later. Let $\widetilde{h}^{(t),i}$ denote the hidden state generated from $E^{(1:t),i}$. We then consider

$$\mathcal{X}^{(t)} = \{-\mathcal{L}_{\mathrm{s}}([E^{(1:t),i}, w_j]), j = 1, ..., V, \ i = 1, ..., k\},$$

where $\mathcal{L}_{\mathrm{s}}(\cdot)$ is defined analogously to (7), and $w_j \in \mathbb{R}^D$ is the embedding of token $z_j$.

Recall that $\epsilon$ is the smoothing parameter. We then apply the sorted SOFT top-$k$ operator to $\mathcal{X}^{(t)}$ to obtain $\{E^{(1:t+1),\ell}\}_{\ell=1}^k$, which are $k$ sequences with the largest likelihoods. More precisely, the sorted SOFT top-$k$ operator yields an output tensor $A^{(t),\epsilon} \in \mathbb{R}^{V \times k \times k}$, where $A_{ji,\ell}^{(t),\epsilon}$ denotes the smoothed indicator of whether $[E^{(1:t),i}, w_j]$ has a rank $\ell$. We then obtain

$$E^{(1:t+1),\ell} = \left[ E^{(1:t),r}, \sum_{j=1}^V \sum_{i=1}^k A_{ji,\ell}^{(t),\epsilon} w_j \right], \tag{9}$$

where $r$ denotes the index $i$ (for $E^{(1:t),i}$'s) associated with the index $\ell$ (for $E^{(1:t+1),\ell}$'s). This is why we use vector representations instead of tokens: this allows us to compute $E^{(t+1),\ell}$ as a weighted sum of all the word embeddings $[w_j]_{j=1}^V$, instead of discarding the un-selected words.

Accordingly, we generate the $k$ hidden states for the $(t+1)$-th decoding step:

$$\widetilde{h}^{(t),\ell} = \sum_{j=1}^{V} \sum_{i=1}^{k} A_{ji,\ell}^{(t),\epsilon} h^{(t),i}, \qquad (10)$$

where $h^{(t),i}$ is the hidden state generated by the decoder based on $E^{(1:t),i}$.

After decoding, we select the sequence with largest likelihood $E^{(1:T),*}$, and maximize the likelihood as follows,

$$\mathcal{L}_{\mathrm{SOFT}} = \prod_{t=1}^{T} \mathbb{P}(y^{(t)} = \bar{y}^{(t)} | \widetilde{h}^{(t\text{-}1),*}(E^{(1:t\text{-}1),*})).$$

We provide the sketch of training procedure in Algorithm 2, where we denote $\mathrm{logit}^{(t),i}$ as $[\log \mathbb{P}(y^{(t+1)} = \omega_j | \widetilde{h}^{(t),i}(E^{(1:t),i}))]_{j=1}^{V}$, which is part of the output of the decoder. More technical details (e.g., backtracking algorithm for finding the index $r$ in (9)) are provided in Appendix C.

Note that integrating the beam search into training essentially yields a very large search space for the model, which is not necessarily affordable sometimes. To alleviate this issue, we further propose a hybrid approach by combining the teacher forcing training with beam search-type training. Specifically, we maximize the weighted likelihood defined as follows,

$$\mathcal{L}_{\mathrm{final}} = \rho \mathcal{L}_{\mathrm{tf}} + (1 - \rho)\mathcal{L}_{\mathrm{SOFT}},$$

where $\rho \in (0, 1)$ is referred to as the "teaching forcing ratio". The teaching forcing loss $\mathcal{L}_{\mathrm{tf}}$ can help reduce the search space and improve the overall performance.

### 5.3 Experiment

We evaluate our proposed beam search + sorted SOFT top-$k$ training procedure using WMT2014 English-French dataset. We adopt beam size 5, teacher forcing ratio $\rho = 0.8$, and $\epsilon = 10^{-1}$. For detailed settings of the training procedure, please refer to Appendix C.

We reproduce the experiment in Bahdanau et al. (2014), and run our proposed training procedure with the identical data pre-processing procedure and the LSTM-based sequence-to-sequence model. Different from Bahdanau et al. (2014), here we also preprocess the data with *byte pair encoding* (Sennrich et al., 2015).

---

**Algorithm 2** Beam search training with SOFT Top-$k$

---

**Require:** Input sequence $s$, target sequence $\bar{y}$; embedding matrix $W \in \mathbb{R}^{V \times D}$; max length $T$; $k$; regularization coefficient $\epsilon$; number of Sinkhorn iteration $L$
$\widetilde{h}_i^{(1)} = h_{\mathrm{e}} = \mathrm{Encoder}(s)$, $E^{(1),i} = w_{\mathrm{s}}$
**for** $t = 1, \cdots, T - 1$ **do**
   **for** $i = 1, \cdots, k$ **do**
      $\mathrm{logit}^{(t),i}, h^{(t),i} = \mathrm{Decoder}(E^{(t),i}, \widetilde{h}^{(t),i})$
      $\log \mathcal{L}_{\mathrm{s}}([E^{(1:t),i}, w_j]) = \log \mathcal{L}_{\mathrm{s}}(E^{(1:t),i}) + \mathrm{logit}_j^{(t),i}$
      $\mathcal{X}^{(t)} = \{-\log \mathcal{L}_{\mathrm{s}}([E^{(1:t),i}, w_j]) \mid j = 1, \cdots, V\}$
   **end for**
   $A^{(t),\epsilon} = \mathrm{Sorted\text{-}SOFT\text{-}Top\text{-}}k(\mathcal{X}^{(t)}, k, \epsilon, L)$
   Compute $E^{(t+1),\ell}, \widetilde{h}^{(t+1),\ell}$ as in (9) and (10)
**end for**
Compute $\nabla \mathcal{L}_{\mathrm{SOFT}}$ and update the model

---

**Results.** As shown in Table 2, the proposed SOFT beam search training procedure achieves an improvement in BLEU score of approximately 0.9. We also include other LSTM-based models for baseline comparison.

**Ablation study.** We replace the SOFT top-$k$ operator with a vanilla top-$k$ operator, i.e., we ignore the gradient of the top-$k$ operation. The obtained BLEU score is 35.84, which suggest a) our SOFT top-$k$ operator and b) incorporating beam search into training both contribute to the improved performance.

## 6 Related Work

We parameterize the top-$k$ operator as an optimal transport problem, which shares the same spirit as Cuturi et al. (2019). Specifically, Cuturi et al. (2019) formulate the ranking and sorting problems as OT problems. Ranking is more complicated than identifying the top-$k$ elements, since one needs to align different ranks to corresponding elements. Therefore, the algorithm complexity per iteration for ranking whole $n$ elements is $\mathcal{O}(n^2)$. Cuturi et al. (2019) also propose an OT problem for finding the $\tau$-quantile in a set of $n$ elements and the algorithm complexity reduces to $\mathcal{O}(n)$. Top-$k$ operator

essentially finds all the elements more extreme than the $(n-k)/n$-quantile, and our proposed algorithm achieves the same complexity $\mathcal{O}(n)$ per iteration. The difference is that top-$k$ operator returns the top-$k$ elements in a given input set, while finding a quantile only yields a certain threshold.

Gumbel-Softmax trick (Jang et al., 2016) can also be utilized to derive a continuous relaxation of the top-$k$ operator. Specifically, Kool et al. (2019) adapted such a trick to sample $k$ elements from $n$ choices, and Xie and Ermon (2019) further applied the trick to stochastic $k$NN, where neural networks are used to approximating the sorting operator. However, as shown in our experiments (see Table 1), the performance of stochastic $k$NN is not as good as deterministic $k$NN.

**Table 2:** BLEU on WMT'14 with single LSTM.

| Algorithm | BLEU |
|---|---|
| Luong et al. (2014) | 33.10 |
| Durrani et al. (2014) | 30.82 |
| Cho et al. (2014) | 34.54 |
| Sutskever et al. (2014) | 30.59 |
| Bahdanau et al. (2014) | 28.45 |
| Jean et al. (2014) | 34.60 |
| Bahdanau et al. (2014) (Our implementation) | 35.38 |
| **Beam Search + Sorted SOFT Top-k** | **36.27** |

Our SOFT beam search training procedure is inspired by several works that incorporate some of the characteristics of beam search into the training procedure (Wiseman and Rush, 2016; Goyal et al., 2018; Bengio et al., 2015). Specifically, Wiseman and Rush (2016) and Goyal et al. (2018) both address the exposure bias issue in beam search. Wiseman and Rush (2016) propose a new loss function in terms of the error made during beam search. This mitigates the misalignment of training and testing in beam search. Later, Goyal et al. (2018) approximates the top-$k$ operator using $k$ softmax operations (This method is described and compared to our proposed method in 4). Such an approximation allows an end-to-end training of beam search. Besides, our proposed training loss $\mathcal{L}_{\text{final}}$ is inspired by Bengio et al. (2015), which combines teacher forcing training procedure and greedy decoding, i.e., beam search with beam size 1.

## 7 Discussion

**Relation to automatic differentiation.** We compute the Jacobian matrix of SOFT top-$k$ operator directly in the backward pass. The OT plan can be obtained by the Sinkhorn algorithm (Algorithm 1), which is iterative and each iteration only involves multiplication and addition. Therefore, we can also apply automatic differentiation (auto-diff) to compute the Jacobian matrix. Specifically, we denote $\Gamma_\ell$ as the transport plan at the $t$-th iteration of Sinkhorn algorithm. The update of $\Gamma_\ell$ can be written as $\Gamma_{\ell+1} = \mathcal{T}(\Gamma_\ell)$, where $\mathcal{T}$ denotes the update of the Sinkhorn algorithm. In order to apply auto-diff, we need to store all the intermediate states, e.g., $p, q, G$ in each iteration, as defined in Algorithm 1 at each iteration. This requires a huge memory size proportional to the number of iterations of the algorithm. In contrast, our backward pass allows us to save memory.

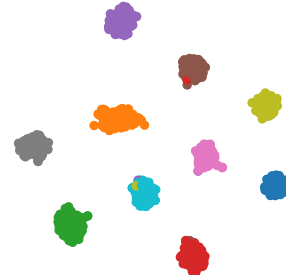

**Figure 5:** Visualization of MNIST data based on features extracted by the neural network-based $k$-NN classifier trained by our proposed method in Section 4.

**Bias and regularization trade-off.** Theorem 2 suggests a trade-off between the regularization and bias of SOFT top-$k$ operator. Specifically, a large $\epsilon$ has a strong smoothing effect on the entropic OT problem, and the corresponding entries of the Jacobian matrix are neither too large nor too small. This eases the end-to-end training process. However, the bias of SOFT top-$k$ operator is large, which can deteriorate the model performance. On the contrary, a smaller $\epsilon$ ensures a smaller bias. Yet the SOFT top-$k$ operator is less smooth, which in turn makes the end-to-end training less efficient.

On the other hand, the bias of SOFT top-$k$ operator also depends on the gap between $x_{\sigma_{k+1}}$ and $x_{\sigma_k}$. In fact, such a gap can be viewed as the signal strength of the problem. A large gap implies that the top-$k$ elements are clearly distinguished from the rest of the elements. Therefore, the bias is expected to be small since the problem is relatively easy. Moreover, in real applications such as neural network-based $k$NN classification, the end-to-end training process promotes neural networks to extract features that exhibit a large gap (as illustrated in Figure 5). Hence, the bias of SOFT top-$k$ operator can be well controlled in practice.

## 8 Broader Impact

This paper makes a significant contribution to extending the frontier of the end-to-end training of compositional models. To the best of our knowledge, our method is the first work targeting at efficient end-to-end training with top-$k$ operation.

We remark that our proposed SOFT top-$k$ operator can be integrated into many existing machine learning methods, and has a great potential to become a standard routine in various applications such as computer vision, natural language processing, healthcare, and computational social science.

## Acknowledgement

We thank Marco Cuturi and Jean-Philippe Vert who provided insight and expertise that greatly assisted the research. We are also grateful to Kihyuk Sohn for comments that greatly improved our earlier version of the manuscript. We thank the anonymous reviewers for their careful reading of our manuscript and their many insightful comments and suggestions.

## Footnotes

[3]Throughout the rest of the paper, we refer to the top-$k$ operator as the top-$k$ operation.

[4]Our implementation is based on github.com/pytorch/vision.git

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
