[Supplementary Material]

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

## A Theoretical Guarantees

**Proposition 1.** Consider the setup in the previous paragraph. Without loss of generality, we assume $\mathcal{X}$ has no duplicates. Then the optimal transport plan $\Gamma^*$ of (1) is

$$\Gamma^*_{\sigma_i,1} = \begin{cases} 1/n, & \text{if } i \le k, \\ 0, & \text{if } k+1 \le i \le n. \end{cases}, \quad \Gamma^*_{\sigma_i,2} = \begin{cases} 0, & \text{if } i \le k, \\ 1/n, & \text{if } k+1 \le i \le n, \end{cases} \tag{11}$$

with $\sigma$ being the sorting permutation, i.e., $x_{\sigma_1} < x_{\sigma_2} < \cdots < x_{\sigma_n}$. Moreover, we have

$$A = n\Gamma^* \cdot [1,0]^\top. \tag{12}$$

*Proof.* We expand the objective function of (1) as

$$\langle C, \Gamma \rangle = \sum_{i=1}^n \left( (x_i - 0)^2 \Gamma_{i,1} + (x_i - 1)^2 \Gamma_{i,2} \right) = \frac{1}{n} \sum_{i=1}^n x_i^2 + \frac{n-k}{n} - 2 \sum_{i=1}^n x_i \Gamma_{i,2}.$$

Therefore, to minimize $\langle C, \Gamma \rangle$, it suffices to maximize $\sum_{i=1}^n x_i \Gamma_{i,2}$. It is straightforward to check

$$\sum_{i=1}^n \Gamma_{i,2} = \frac{n-k}{n} \quad \text{and} \quad \Gamma_{i,2} \le \frac{1}{n}$$

for any $i = 1, \ldots, n$. Hence, maximizing $\sum_{i=1}^n x_i \Gamma_{i,2}$ is essentially selecting the largest $n - K$ elements from $\mathcal{X}$, and the maximum is attained at

$$\Gamma^*_{\sigma_i,2} = \begin{cases} 0, & \text{if } i \le k, \\ 1/n, & \text{if } k+1 \le i \le n. \end{cases}$$

The constraint $\Gamma \mathbf{1}_m = \mu$ further implies that $\Gamma^*_{i,1}$ satisfies (11). Thus, $A$ can be parameterized as $A = n\Gamma^* \cdot [1,0]^\top$. □

We then show that after adding entropy regularization the problem is differentiable.

**Theorem 1.** For any $\epsilon > 0$, SOFT top-$k$ operator: $\mathcal{X} \mapsto A^\epsilon$ is differentiable, as long as the cost $C_{ij}$ is differentiable with respect to $x_i$ for any $i, j$. Moreover, the Jacobian matrix of SOFT top-$k$ operator always has a nonzero entry for any $\mathcal{X} \in \mathbb{R}^n$.

*Proof.* We first prove the differentiability. This part of proof mirrors the proof in Luise et al. (2018). By Sinkhorn's scaling theorem,

$$\Gamma^{*,\epsilon} = \text{diag}(e^{\frac{\xi^*}{\epsilon}}) e^{-\frac{C}{\epsilon}} \text{diag}(e^{\frac{\zeta^*}{\epsilon}}).$$

Therefore, since $C_{ij}$ is differentiable, $\Gamma^{*,\epsilon}$ is differentiable if $(\xi^*, \zeta^*)$ is differentiable as a function of input scores $X$.

Let us set

$$\mathcal{L}(\xi, \zeta; \mu, \nu, C) = \xi^T \mu + \zeta^T \nu - \epsilon \sum_{i,j=1}^{n,m} e^{-\frac{C_{ij} - \xi_i - \zeta_j}{\epsilon}}.$$

and recall that $(\xi^*, \zeta^*) = \text{argmax}_{\xi,\zeta} L(\xi, \zeta; \mu, \nu, C)$. The differentiability of $(\xi^*, \zeta^*)$ is proved using the Implicit Function theorem and follows from the differentiability and strong convexity in $(\xi^*, \zeta^*)$ of the function $\mathcal{L}$.

Now we prove that $dA^\epsilon/dx_\ell$ always has a nonzero entry for $l = 1, \cdots, n$. First, we prove that for any $\ell \in \{1, \cdots, n\}$, $d\Gamma^{*,\epsilon}/dx_\ell$ always has a nonzero entry. We will prove it by contradiction. Specifically, the KKT conditions for the stationarity are as follows

$$\xi_i^* + \zeta_j^* = (x_i - y_j)^2 - \epsilon \log \Gamma_{ij}^{*,\epsilon}, \quad \forall i = 1, \cdots, n, j = 1, \cdots, m.$$

If we view the above formula as a linear equation set of the dual variables, it has $nm$ equations and $m + n$ variables. Therefore, there are $nm - m - n$ redundant equations. Suppose one of the scores $x_\ell$ has an infinitesimal change $\delta x_\ell$. Assuming $\Gamma^{*,\epsilon}$ does not change, we have a new set of linear equations,

$$\xi_i^* + \zeta_j^* = (x_i - y_j)^2 - \epsilon \log \Gamma_{ij}^{*,\epsilon}, \quad \forall i \ne \ell,$$
$$\xi_\ell^* + \zeta_j^* = (x_\ell + \delta x_\ell - y_j)^2 + \delta C_{\ell j} - \epsilon \log \Gamma_{\ell j}^{*,\epsilon}.$$

Easy to verify that this set of linear equations has no solution. Therefore, there must be at least one entry in $\Gamma^{*,\epsilon}$ has changed. As a result, $d\Gamma^{*,\epsilon}/dx_\ell$ always has a nonzero entry. We denote this entry as $\Gamma^{*,\epsilon}_{i'j'}$. Since $\Gamma^{*,\epsilon}_{i'j'} + \Gamma^{*,\epsilon}_{i',3-j'} = \mu_{i'}$, we have

$$\frac{d\Gamma^{*,\epsilon}_{i',3-j'}}{dx_\ell} = -\frac{d\Gamma^{*,\epsilon}_{i'j'}}{dx_\ell} \neq 0.$$

Therefore, there must be a nonzero entry in the first column of $d\Gamma^{*,\epsilon}/dx_\ell$. Recall $A^\epsilon$ is the first column of $\Gamma^{*,\epsilon}$. As a result, there must be a nonzero entry in $dA^\epsilon/dx_\ell$ for any $\ell \in \{1, \cdots, n\}$.

$\square$

Second, we would like to know after smoothness relaxation, how much bias is introduced to $A^\epsilon$.

**Lemma 1.** Denote the feasible set of optimal transport problem as $\Delta = \{\Gamma : \Gamma \in [0,1]^{n \times m}, \Gamma \mathbf{1}_m = \mu, \Gamma \mathbf{1}_n = \nu\}$. Assume the optimal transport plan is unique. Denote $\Gamma^*$ as the optimal transport plan,

$$\Gamma^* = \underset{\Gamma \in \Delta}{\operatorname{argmin}} f(\Gamma) = \underset{\Gamma \in \Delta}{\operatorname{argmin}} \langle C, \Gamma \rangle,$$

and $\Gamma^{*,\epsilon}$ as the entropy regularized transport plan,

$$\Gamma^{*,\epsilon} = \underset{\Gamma \in \Delta}{\operatorname{argmin}} f^\epsilon(\Gamma) = \underset{\Gamma \in \Delta}{\operatorname{argmin}} f(\Gamma) - \epsilon H(\Gamma) = \underset{\Gamma \in \Delta}{\operatorname{argmin}} \langle C, \Gamma \rangle + \epsilon \sum_{i,j} \Gamma_{ij} \ln \Gamma_{ij}.$$

We can bound the difference between $\Gamma^*$ and $\Gamma^{*,\epsilon}$ to be

$$\|\Gamma^* - \Gamma^{*,\epsilon}\|_F \leq \epsilon \frac{(\ln n + \ln m)}{B},$$

where $\|\cdot\|_F$ is the Frobenius norm, and $B$ is a positive constant irrelevant to $\epsilon$.

*Proof.* Note that $H(\Gamma)$ is the entropy function. Since $0 \leq \Gamma_{ij} \leq 1$ and $\sum_{ij} \Gamma_{ij} = 1$ for any $\Gamma \in \Delta$, we can view $\Delta$ as the subset of a simplex. Therefore,

1. $H(\Gamma)$ is non-negative.

2. The maximum of $H(\Gamma)$ in the simplex can be obtained at $\Gamma_{ij} \equiv \frac{1}{nm}$. Therefore the maximum value is $(\ln n + \ln m)$.

Therefore, $0 \leq H(\Gamma) \leq (\ln n + \ln m)$ for any $\Gamma \in \Delta$.

Since $H(\Gamma) \geq 0$, we have $f^\epsilon(\Gamma) \leq f(\Gamma)$ for any $\Gamma \in \Delta$. As a result, we have $f^\epsilon(\Gamma^{*,\epsilon}) \leq f(\Gamma^*)$. In other words, we have

$$\langle C, \Gamma^{*,\epsilon} \rangle - \epsilon H(\Gamma^{*,\epsilon}) - \langle C, \Gamma^* \rangle \leq 0.$$

Therefore,

$$\langle C, \Gamma^{*,\epsilon} - \Gamma^* \rangle = \langle C, \Gamma^{*,\epsilon} \rangle - \langle C, \Gamma^* \rangle \leq \epsilon H(\Gamma^{*,\epsilon}) \leq \epsilon(\ln n + \ln m).$$

Since the optimal transport problem is a linear optimization problem, $\Gamma^*$ is one of the vertices of $\Delta$. Denote $e_0, e_1, \cdots, e_J$ as the vertices of $\Delta$, and without loss of generality we assume $e_0 = \Gamma^*$. Since $\Gamma^{*,\epsilon} \in \Delta$, we can denote $\Gamma^{*,\epsilon} = \sum_{j=0}^J \lambda_j e_j$, where $\lambda_j \geq 0$, and $\sum_j \lambda_j = 1$. Since $\Gamma^*$ is unique, we have

$$\langle C, e_j - e_0 \rangle > 0, \quad \forall j = 1, \cdots, J.$$

Denote $B_j = \langle C, e_j - e_0 \rangle$. Since the space we are considering is Euclidean space (if we reshape the matrices into vectors), we can write the inner product as

$$B_j = \langle C, e_j - e_0 \rangle = \|C\|_F \|e_j - e_0\|_F \cos \theta_{(C, e_j - e_0)} > 0.$$

So we have $\cos \theta_{(C, e_j - e_0)} > 0$. In other words, the angle between $C$ and $e_j - e_0$ is always smaller than $\frac{\pi}{2}$. Therefore, the angle between $C$ and the affine combination of $e_j - e_0$, namely $\sum_{j=0}^J \lambda_j(e_j - e_0)$, is also smaller than $\frac{\pi}{2}$. More specifically, we have

$$\cos \theta_{(C, \Gamma^{*,\epsilon} - \Gamma^*)} = \cos \theta_{(C, \sum_{j=0}^J \lambda_j(e_j - e_0))} \geq \min_j \cos \theta_{(C, e_j - e_0)} = \min_j \frac{B_j}{\|C\|_F \|e_j - e_0\|_F}.$$

Therefore, we have

$$\|\Gamma^{*,\epsilon} - \Gamma^*\|_F = \frac{\langle C, \Gamma^{*,\epsilon} - \Gamma^* \rangle}{\|C\|_F \cos \theta_{(C, \Gamma^{*,\epsilon} - \Gamma^*)}} \leq \frac{\epsilon(\ln n + \ln m)}{\|C\|_F \min_j \frac{B_j}{\|C\|_F \|e_j - e_0\|_F}} = \frac{\epsilon(\ln n + \ln m)}{\min_j \frac{B_j}{\|e_j - e_0\|_F}}.$$

Denote $B = \min_j \frac{B_j}{\|e_j - e_0\|_F}$, and we have the conclusion. $\square$

**Remark 1.** In Theorem 1 we restricted the optimal solution to be unique, only for clarity purpose. If it is not unique, similar conclusion holds, except that the proof is more tedious – instead of divide the vertices into $e_0$ and others, we need to divide it into the vertices that are optimal solutions and the others.

**Lemma 2.** At each of the vertices of $\Delta$, the entries of $\Gamma$ are either $0$ or $1/n$ for $\Gamma \in \Delta$.

*Proof.* The key idea is to prove by contradiction: If there exist $i, j$ such that $\Gamma_{ij} \in (0, 1/n)$, then $\Gamma$ cannot be a vertex.

To ease the discussion, we denote $Z = n\Gamma$. We will first prove that the entries of $Z$ are either $0$ or $1$ at the vertices.

Notice that

$$Z_{i,1} + Z_{i,2} = 1, \quad \forall i = 1, \cdots, n,$$
$$\sum_i Z_{i,1} = k,$$
$$\sum_i Z_{i,2} = n - k.$$

If there exists an entry $Z_{i',j'} \in (0, 1)$, then

1. $Z_{i',3-j'} \in (0, 1)$.

2. there must exist $i'' \neq i'$, such that $Z_{i'',j'} \in (0, 1)$. This is because $\sum_{i=1}^{n} Z_{i,j}$ is an integer, and $Z_{i',j'}$ is not.

3. As a result, $Z_{i'',3-j'} \in (0, 1)$.

Therefore, consider $\delta \in (-\min\{1 - Z_{i',j'}, Z_{i',j'}\}, \min\{1 - Z_{i',j'}, Z_{i',j'}\})$ and denote

$$\widetilde{Z}_{ij}^{(1)} = \begin{cases} Z_{i',j'} + \delta, & \text{if } i = i', j = j', \\ Z_{i',3-j'} - \delta, & \text{if } i = i', j = 3 - j', \\ Z_{i'',j'} - \delta, & \text{if } i = i'', j = j', \\ Z_{i'',3-j'} + \delta, & \text{if } i = i'', j = 3 - j', \\ Z_{i,j}, & \text{otherwise.} \end{cases}$$

$$\widetilde{Z}_{ij}^{(2)} = \begin{cases} Z_{i',j'} - \delta, & \text{if } i = i', j = j', \\ Z_{i',3-j'} + \delta, & \text{if } i = i', j = 3 - j', \\ Z_{i'',j'} + \delta, & \text{if } i = i'', j = j', \\ Z_{i'',3-j'} - \delta, & \text{if } i = i'', j = 3 - j', \\ Z_{i,j}, & \text{otherwise.} \end{cases}$$

We can easily verify that $\widetilde{Z}^{(1)}/n, \widetilde{Z}^{(2)}/n \in \Delta$, and also $Z = (\widetilde{Z}^{(1)} + \widetilde{Z}^{(2)})/2$. Therefore, $Z$ cannot be a vertex.

$\square$

**Lemma 3.** Given a set of scalar $\{x_1, \cdots, x_n\}$, we sort it to be $\{x_{\sigma_1}, \cdots, x_{\sigma_n}\}$. If Euclidean square cost is adopted, $\Gamma^*$ has the following form,

$$\Gamma_{ij}^* = \begin{cases} 1/n, & \text{if } i = \sigma_\ell, j = 1, \ell \leq k \\ 0, & \text{if } i = \sigma_\ell, j = 1, k < \ell \leq n \\ 1/n, & \text{if } i = \sigma_\ell, j = 2, k < \ell \leq n \\ 0, & \text{if } i = \sigma_\ell, j = 2, \ell \leq k \end{cases}$$

And $\min_j \frac{B_j}{\|e_j - e_0\|_F}$ is attained at at a vertex $\Gamma^{**}$, where $\Gamma_{ij}^{**} = \Gamma_{ij}^*$ except that the $\sigma_k$-th row and the $\sigma_{k+1}$-th row are swapped. As a result, we have

$$\min_j \frac{B_j}{\|e_j - e_0\|_F} = n(x_{\sigma_{k+1}} - x_{\sigma_k}).$$

*Proof.* From Lemma 2, in each vertex the entries of $\Gamma$ is either $0$ or $1/n$. Also, $\Gamma^* \in \Delta = \{\Gamma : \Gamma \in [0,1]^{n\times m}, \Gamma \mathbf{1}_m = \mathbf{1}_n/n, \Gamma \mathbf{1}_n = [k/n, (n-k)/n]^\top\}$. Therefore, for the $j$-th vertex, there are $k$ entries with value $1/n$ in the first row of $\Gamma$. Denote the row indices of these $k$ entries as $\mathcal{I}_j$, and $\Omega = \{1, \cdots, n\}$. Then for each vertex we have

$$\Gamma_{i,1} = 1/n, \quad \forall i \in \mathcal{I}_j$$
$$\Gamma_{i,1} = 0, \quad \forall i \in \Omega\backslash\mathcal{I}_j$$
$$\Gamma_{i,2} = 1/n, \quad \forall i \in \Omega\backslash\mathcal{I}_j$$
$$\Gamma_{i,2} = 0, \quad \forall i \in \mathcal{I}_j.$$

Denote $\mathcal{I}^* = \{\sigma_1, \cdots, \sigma_k\}$. We now prove that $\mathcal{I}^*$ corresponds to the optimal solution $\Gamma^*$. This is because for any $j \in \{1, \cdots, J\}$

$$\Gamma(\mathcal{I}_j) - \Gamma(\mathcal{I}^*) = \left(\sum_{i\in\mathcal{I}_j} x_i^2 + \sum_{i\in\Omega\backslash\mathcal{I}_j} (x_i-1)^2\right) - \left(\sum_{i\in\mathcal{I}^*} x_i^2 + \sum_{i\in\Omega\backslash\mathcal{I}^*} (x_i-1)^2\right)$$

$$= \left(\sum_{i\in\Omega} x_i^2 - \sum_{i\in\Omega\backslash\mathcal{I}_j} 2x_i + (n-k)\right) - \left(\sum_{i\in\Omega} x_i^2 - \sum_{i\in\Omega\backslash\mathcal{I}^*} 2x_i + (n-k)\right)$$

$$= 2\left(\sum_{i\in\Omega\backslash\mathcal{I}^*} x_i - \sum_{i\in\Omega\backslash\mathcal{I}_j} x_i\right) \geq 0,$$

where the last step is because the elements with indices $\Omega\backslash\mathcal{I}_j$ is the largest $n-k$ elements. Therefore we have $\Gamma(\mathcal{I}^*) = \Gamma^*$.

Now let's compute $\min_{j\neq 0} B_j/\|e_j - e_0\|$. Denote set subtraction $\mathcal{A} - \mathcal{B}$ as the set if elements that belongs to $\mathcal{A}$ but do not belong to $\mathcal{B}$, and $|\mathcal{A}|$ as the number of elements in $\mathcal{A}$.

$$\frac{B_j}{\|e_j - e_0\|} = \frac{B_j}{\|\Gamma(\mathcal{I}_j) - \Gamma(\mathcal{I}^*)\|}$$

$$= 2\frac{\sum_{i\in\Omega\backslash\mathcal{I}^*} x_i - \sum_{i\in\Omega\backslash\mathcal{I}_j} x_i}{2\sqrt{|\mathcal{I}^* - \mathcal{I}_j|}/n}$$

$$= n\frac{\sum_{i\in(\mathcal{I}_j-\mathcal{I}^*)} x_i - \sum_{i\in(\mathcal{I}^*-\mathcal{I}_j)} x_i}{\sqrt{|\mathcal{I}^* - \mathcal{I}_j|}},$$

where the second line can be obtained by substituting the definition of $B_j$. Notice that $\mathcal{I}_j - \mathcal{I}^* \in \Omega\backslash\mathcal{I}^*$ and $\mathcal{I}^* - \mathcal{I}_j \in \mathcal{I}^*$. Any element with index in $\Omega\backslash\mathcal{I}^*$ is larger than any element in $\mathcal{I}^*$ by at least $x_{\sigma_{K+1}} - x_{\sigma_K}$. Then we have

$$\frac{B_j}{\|e_j - e_0\|} = N\frac{\sum_{i\in(\mathcal{I}_j-\mathcal{I}^*)} x_i - \sum_{i\in(\mathcal{I}^*-\mathcal{I}_j)} x_i}{\sqrt{|\mathcal{I}^* - \mathcal{I}_j|}}$$

$$\geq N\frac{|\mathcal{I}^* - \mathcal{I}_j|(x_{\sigma_{K+1}} - x_{\sigma_K})}{\sqrt{|\mathcal{I}^* - \mathcal{I}_j|}}$$

$$\geq N(x_{\sigma_{K+1}} - x_{\sigma_K}),$$

where the last step is because for $j \neq 0$, $|\mathcal{I}^* - \mathcal{I}_j|$ is at least 1.

Also notice that the value $n(x_{\sigma_{k+1}} - x_{\sigma_k})$ can be attained at $\mathcal{I}_{j^*} = \{\sigma_1, \cdots, \sigma_{k-1}, \sigma_{k+1}\}$. Therefore we have

$$\min_j \frac{B_j}{\|e_j - e_0\|} = n(x_{\sigma_{k+1}} - x_{\sigma_k}).$$

$\square$

**Theorem 2.** Given a distinct sequence $\mathcal{X}$ and its sorting permutation $\sigma$, with Euclidean square cost function, for the proposed top-$k$ solver we have

$$\|\Gamma^{*,\epsilon} - \Gamma^*\| \leq \frac{\epsilon(\ln n + \ln 2)}{n(x_{\sigma_{k+1}} - x_{\sigma_k})}.$$

*Proof.* This is a direct conclusion with Lemma 1 and Lemma 3. $\square$

# B   The Expression of the Gradient of $A^\epsilon$

In this section we will derive the expression of $dA^\epsilon/dx_i$. We first list a few reminders that will be used later:

- $\{x_i\}_{i=1}^n$ is a scalar set to be solved for top-$k$. $\{y_j\}_{j=1}^m$ is taken to be $\{0,1\}$.

- $C \in \mathbb{R}^{n \times m}$ is the cost matrix, usually defined as $C_{ij} = (x_i - y_j)^2$.

- The loss function of entropic optimal transport is

$$\Gamma^{*,\epsilon} = \operatorname*{argmin}_{\Gamma \in \Delta} f^\epsilon(\Gamma) = \operatorname*{argmin}_{\Gamma \in \Delta} \langle C, \Gamma \rangle + \epsilon \sum_{i,j} \Gamma_{ij} \ln \Gamma_{ij},$$

  where $\Delta = \{\Gamma : \Gamma \in [0,1]^{n \times m}, \Gamma \mathbf{1}_m = \mu, \Gamma \mathbf{1}_n = \nu\}$.

- The dual problem of the above optimization problem is

$$\xi^*, \zeta^* = \operatorname*{argmax}_{\xi, \zeta} \mathcal{L}(\xi, \zeta; C),$$

  where

$$\mathcal{L}(\xi, \zeta; C) = \xi^\top \mu + \zeta^\top \nu - \epsilon \sum_{i,j=1}^{n,m} e^{-\frac{C_{ij} - \xi_i - \zeta_j}{\epsilon}}.$$

  And it is connected to the prime form by

$$\Gamma^{*,\epsilon} = \operatorname{diag}(e^{\frac{\xi^*}{\epsilon}}) e^{-\frac{C}{\epsilon}} \operatorname{diag}(e^{\frac{\zeta^*}{\epsilon}}).$$

  The converged $p, q$ in Algorithm 1 is actually $e^{\frac{\xi^*}{\epsilon}}$ and $e^{\frac{\zeta^*}{\epsilon}}$.

If we obtain the expression for $\frac{d\xi^*}{dC}$ and $\frac{d\zeta^*}{dC}$, we can obtain the expression for $\frac{dA^\epsilon}{dx_i}$.

In this section only, we denote $\Gamma = \Gamma^{*,\epsilon}$, to shorten the notation. The multiplication of 3rd-order tensors mirrors the multiplication of matrices: we always use the last dimension of the first input to multiplies the first dimension of the second input. We denote $\bar{b} = b_{:-1}$ as $b$ removing the last entry, $\bar{\nu} = \nu_{:-1}$ as $\nu$ removing the last entry, $\bar{\Gamma} = \Gamma_{:,:-1}$ as $\Gamma$ removing the last column.

**Theorem 3.** $\frac{d\xi^*}{dC}$ and $\frac{d\zeta^*}{dC}$ have the following expression,

$$\begin{bmatrix} \frac{d\xi^*}{dC} \\ \frac{d\zeta^*}{dC} \end{bmatrix} = \begin{bmatrix} -H^{-1}D \\ \mathbf{0} \end{bmatrix}$$

where $-H^{-1}D \in \mathbb{R}^{(n+m-1) \times n \times m}, \mathbf{0} \in \mathbb{R}^{1 \times n \times m}$, and

$$D_{\ell ij} = \frac{1}{\epsilon} \begin{cases} \delta_{\ell i} \Gamma_{ij}, \ell = 1, \cdots, n \\ \delta_{\ell j} \Gamma_{ij}, \ell = n+1, \cdots, n+m-1 \end{cases}$$

$$H^{-1} = -\epsilon \begin{bmatrix} (\operatorname{diag}(\mu))^{-1} + (\operatorname{diag}(\mu))^{-1} \bar{\Gamma} \mathcal{K}^{-1} \bar{\Gamma}^T (\operatorname{diag}(\mu))^{-1} & -(\operatorname{diag}(\mu))^{-1} \bar{\Gamma} \mathcal{K}^{-1} \\ -\mathcal{K}^{-1} \bar{\Gamma}^T (\operatorname{diag}(\mu))^{-1} & \mathcal{K}^{-1} \end{bmatrix}$$

$$\mathcal{K} = \operatorname{diag}(\bar{\nu}) - \bar{\Gamma}^T (\operatorname{diag}(\mu))^{-1} \bar{\Gamma}.$$

*Proof.* Notice that there is one redundant dual variable, since $\mu \mathbf{1}_n = \nu \mathbf{1}_m = 1$. Therefore, we can rewrite $\mathcal{L}(\xi, \zeta; C)$ as

$$\mathcal{L}(\xi, \bar{\zeta}; C) = \xi^T \mu + \bar{\zeta}^T \bar{\nu} - \epsilon \sum_{i,j=1}^{n,m-1} e^{\frac{-C_{ij} + \xi_i + \zeta_j}{\epsilon}} - \epsilon \sum_{i=1}^n e^{\frac{-C_{im} + \xi_i}{\epsilon}}.$$

Denote

$$\phi(\xi, \bar{\zeta}, C) = \frac{d\mathcal{L}(\xi, \bar{\zeta}; C)}{d\xi} = \mu - F\mathbf{1}_m, \tag{13}$$

$$\psi(\xi, \bar{\zeta}, C) = \frac{d\mathcal{L}(\xi, \bar{\zeta}; C)}{d\bar{\zeta}} = \bar{\nu} - \bar{F}^\top \mathbf{1}_n, \tag{14}$$

where

$$F_{ij} = e^{\frac{-C_{ij}+\xi_i+\varsigma_j}{\epsilon}}, \quad \forall i = 1, \cdots, n, \quad j = 1, \cdots, m-1$$

$$F_{im} = e^{\frac{-C_{im}+\xi_i}{\epsilon}}, \quad \forall i = 1, \cdots, n,$$

$$\bar{F} = F_{:,:-1}.$$

Since $(\xi^*, \bar{\zeta}^*)$ is a maximum of $\mathcal{L}(\xi, \bar{\zeta}; C)$, we have

$$\phi(\xi^*, \bar{\zeta}^*, C) = 0,$$

$$\psi(\xi^*, \bar{\zeta}^*, C) = 0.$$

Therefore,

$$\frac{d\phi(\xi^*, \bar{\zeta}^*, C)}{dC} = \frac{\partial\phi(\xi^*, \bar{\zeta}^*, C)}{\partial C} + \frac{\partial\phi(\xi^*, \bar{\zeta}^*, C)}{\partial \xi^*}\frac{d\xi^*}{dC} + \frac{\partial\phi(\xi^*, \bar{\zeta}^*, C)}{\partial \bar{\zeta}^*}\frac{d\bar{\zeta}^*}{dC} = 0,$$

$$\frac{d\psi(\xi^*, \bar{\zeta}^*, C)}{dC} = \frac{\partial\psi(\xi^*, \bar{\zeta}^*, C)}{\partial C} + \frac{\partial\psi(\xi^*, \bar{\zeta}^*, C)}{\partial \xi^*}\frac{d\xi^*}{dC} + \frac{\partial\psi(\xi^*, \bar{\zeta}^*, C)}{\partial \bar{\zeta}^*}\frac{d\bar{\zeta}^*}{dC} = 0.$$

Therefore,

$$\begin{bmatrix} \frac{d\xi^*}{dC} \\ \frac{d\bar{\zeta}^*}{dC} \end{bmatrix} = - \begin{bmatrix} \frac{\partial\phi(\xi^*, \bar{\zeta}^*, C)}{\partial \xi^*} & \frac{\partial\phi(\xi^*, \bar{\zeta}^*, C)}{\partial \bar{\zeta}^*} \\ \frac{\partial\psi(\xi^*, \bar{\zeta}^*, C)}{\partial \xi^*} & \frac{\partial\psi(\xi^*, \bar{\zeta}^*, C)}{\partial \bar{\zeta}^*} \end{bmatrix}^{-1} \begin{bmatrix} \frac{\partial\phi(\xi^*, \bar{\zeta}^*, C)}{\partial C} \\ \frac{\partial\psi(\xi^*, \bar{\zeta}^*, C)}{\partial C} \end{bmatrix}$$

$$\triangleq -H^{-1} \begin{bmatrix} D^{(1)} \\ D^{(2)} \end{bmatrix}$$

$$\triangleq -H^{-1}D.$$

Now let's compute each of the terms.

$$\frac{\partial\phi(\xi^*, \bar{\zeta}^*, C)_h}{\partial C_{ij}} = -\frac{\partial[F\mathbf{1}_m]_h}{\partial C_{ij}} = -\frac{\partial}{\partial C_{ij}}\left(\sum_{\ell=1}^{m-1} e^{\frac{-C_{h\ell}+\xi_h+\varsigma_\ell}{\epsilon}} + e^{\frac{-C_{hm}+\xi_h}{\epsilon}}\right)$$

$$= \frac{1}{\epsilon}\delta_{hi}F_{ij} = \frac{1}{\epsilon}\delta_{hi}\Gamma_{ij}$$

$$\forall h = 1, \cdots, n, \quad i = 1, \cdots, n, \quad j = 1, \cdots, m$$

$$\frac{\partial\psi(\xi^*, \bar{\zeta}^*, C)_\ell}{\partial C_{ij}} = -\frac{\partial[\bar{F}^\top \mathbf{1}_n]_\ell}{\partial C_{ij}} = -\frac{\partial}{\partial C_{ij}}\sum_{h=1}^{n} e^{\frac{-C_{h\ell}+\xi_h+\varsigma_\ell}{\epsilon}}$$

$$= \frac{1}{\epsilon}\delta_{\ell j}F_{ij} = \frac{1}{\epsilon}\delta_{\ell j}\Gamma_{ij}$$

$$\forall \ell = 1, \cdots, m-1, \quad i = 1, \cdots, n, \quad j = 1, \cdots, m$$

$$\frac{\partial\phi(\xi^*, \bar{\zeta}^*, C)_h}{\partial \xi_i^*} = -\frac{\partial[F\mathbf{1}_m]_h}{\partial \xi_i^*} = -\frac{\partial}{\partial \xi_i^*}\left(\sum_{\ell=1}^{m-1} e^{\frac{-C_{h\ell}+\xi_h+\varsigma_\ell}{\epsilon}} + e^{\frac{-C_{hm}+\xi_h}{\epsilon}}\right)$$

$$= -\frac{1}{\epsilon}\delta_{hi}\sum_{\ell=1}^{m} F_{h\ell} = -\frac{1}{\epsilon}\delta_{hi}\mu_h$$

$$\forall h = 1, \cdots, n, \quad i = 1, \cdots, n$$

$$\frac{\partial\phi(\xi^*, \bar{\zeta}^*, C)_h}{\partial \bar{\zeta}_j^*} = -\frac{\partial[F\mathbf{1}_m]_h}{\partial \bar{\zeta}_j^*} = -\frac{\partial}{\partial \bar{\zeta}_j^*}\left(\sum_{\ell=1}^{m-1} e^{\frac{-C_{h\ell}+\xi_h+\varsigma_\ell}{\epsilon}} + e^{\frac{-C_{hm}+\xi_h}{\epsilon}}\right)$$

$$= -\frac{1}{\epsilon}\sum_{\ell=1}^{m-1} \delta_{\ell j}F_{h\ell} = -\frac{1}{\epsilon}F_{hj} = -\frac{1}{\epsilon}\Gamma_{hj}$$

$$\forall h = 1, \cdots, n, \quad j = 1, \cdots, m-1$$

$$\frac{\partial\psi(\xi^*, \bar{\zeta}^*, C)_\ell}{\partial \xi_i^*} = -\frac{\partial[\bar{F}^\top \mathbf{1}_n]_\ell}{\partial \xi_i^*} = -\frac{\partial}{\partial \xi_i^*}\sum_{h=1}^{n} e^{\frac{-C_{h\ell}+\xi_h+\varsigma_\ell}{\epsilon}}$$

$$= -\frac{1}{\epsilon}\sum_{h=1}^{n} \delta_{hi}F_{h\ell} = -\frac{1}{\epsilon}F_{i\ell} = -\frac{1}{\epsilon}\Gamma_{i\ell}$$

$$\forall \ell = 1, \cdots, m-1, \quad i = 1, \cdots, n$$

$$\frac{\partial \psi(\xi^*, \bar{\zeta}^*, C)_\ell}{\partial \bar{\zeta}_j^*} = -\frac{\partial [\bar{F}^\top \mathbf{1}_n]_\ell}{\partial \bar{\zeta}_j^*} = -\frac{\partial}{\partial \bar{\zeta}_j^*} \sum_{h=1}^n e^{\frac{-C_{h\ell} + \xi_h + \zeta_\ell}{\epsilon}}$$

$$= -\frac{1}{\epsilon} \sum_{h=1}^n \delta_{\ell j} F_{h\ell} = -\frac{1}{\epsilon} \delta_{\ell j} \nu_\ell$$

$$\forall \ell = 1, \cdots, m-1, \quad j = 1, \cdots, m-1.$$

To sum up, we have

$$H = -\frac{1}{\epsilon} \begin{bmatrix} \mathrm{diag}(\mu) & \bar{\Gamma} \\ \bar{\Gamma}^T & \mathrm{diag}(\bar{\nu}) \end{bmatrix}.$$

Following the formula for inverse of block matrices,

$$\begin{bmatrix} \mathbf{A} & \mathbf{B} \\ \mathbf{C} & \mathbf{D} \end{bmatrix}^{-1} = \begin{bmatrix} \mathbf{A}^{-1} + \mathbf{A}^{-1}\mathbf{B}(\mathbf{D} - \mathbf{C}\mathbf{A}^{-1}\mathbf{B})^{-1}\mathbf{C}\mathbf{A}^{-1} & -\mathbf{A}^{-1}\mathbf{B}(\mathbf{D} - \mathbf{C}\mathbf{A}^{-1}\mathbf{B})^{-1} \\ -(\mathbf{D} - \mathbf{C}\mathbf{A}^{-1}\mathbf{B})^{-1}\mathbf{C}\mathbf{A}^{-1} & (\mathbf{D} - \mathbf{C}\mathbf{A}^{-1}\mathbf{B})^{-1} \end{bmatrix},$$

denote

$$\mathcal{K} = \mathrm{diag}(\bar{\nu}) - \bar{\Gamma}^T (\mathrm{diag}(\mu))^{-1} \bar{\Gamma}.$$

Note that $\mathcal{K}$ is just a scalar for SOFT top-$k$ operator, and is a $(k-1) \times (k-1)$ matrix for sorted SOFT top-$k$ operator. Therefore computing its inverse is not expensive. Finally we have

$$H^{-1} = -\epsilon \begin{bmatrix} (\mathrm{diag}(\mu))^{-1} + (\mathrm{diag}(\mu))^{-1}\bar{\Gamma}\mathcal{K}^{-1}\bar{\Gamma}^T(\mathrm{diag}(\mu))^{-1} & -(\mathrm{diag}(\mu))^{-1}\bar{\Gamma}\mathcal{K}^{-1} \\ -\mathcal{K}^{-1}\bar{\Gamma}^T(\mathrm{diag}(\mu))^{-1} & \mathcal{K}^{-1} \end{bmatrix}.$$

And also

$$D_{hij}^{(1)} = \frac{1}{\epsilon} \delta_{hi} \Gamma_{ij}$$

$$D_{\ell ij}^{(2)} = \frac{1}{\epsilon} \delta_{\ell j} \Gamma_{ij}.$$

The above derivation can actually be viewed as we explicitly force $\zeta_m = 0$, i.e., no matter how $C$ changes, $\zeta_m$ does not change. Therefore, we can treat $\frac{d\zeta_m}{dC} = \mathbf{0}_{n \times m}$, and we get the equation in the theorem. $\qquad \square$

After we obtain $\frac{d\xi^*}{dC}$ and $\frac{d\zeta^*}{dC}$, we can now compute $\frac{d\Gamma}{dC}$.

$$\frac{d\Gamma_{h\ell}}{dC_{ij}} = \frac{d}{dC_{ij}} e^{\frac{-C_{h\ell} + \xi_h^* + \zeta_\ell^*}{\epsilon}} = \frac{1}{\epsilon} \left( -\Gamma_{h\ell} \delta_{ih} \delta_{j\ell} + \Gamma_{h\ell} \frac{d\xi_h^*}{dC_{ij}} + \Gamma_{h\ell} \frac{d\zeta_\ell^*}{dC_{ij}} \right).$$

Finally, in the back-propagation step, we can compute the gradient of the loss $L$ w.r.t. $C$,

$$\frac{dL}{dC_{ij}} = \sum_{h,\ell=1}^{n,m} \frac{dL}{d\Gamma_{h\ell}} \frac{d\Gamma_{h\ell}}{dC_{ij}}$$

$$= \frac{1}{\epsilon} \left( -\sum_{h,\ell=1}^{n,m} \frac{dL}{d\Gamma_{h\ell}} \Gamma_{h\ell} \delta_{in} \delta_{j\ell} + \sum_{h,\ell=1}^{n,m} \frac{dL}{d\Gamma_{h\ell}} \Gamma_{h\ell} \frac{d\xi_h^*}{dC_{ij}} + \sum_{h,\ell=1}^{n,m} \frac{dL}{d\Gamma_{h\ell}} \Gamma_{h\ell} \frac{d\zeta_\ell^*}{dC_{ij}} \right)$$

$$= \frac{1}{\epsilon} \left( -\frac{dL}{d\Gamma_{ij}} \Gamma_{ij} + \sum_{h,\ell=1}^{n,m} \frac{dL}{d\Gamma_{h\ell}} \Gamma_{h\ell} \frac{d\xi_h^*}{dC_{ij}} + \sum_{h,\ell=1}^{n,m} \frac{dL}{d\Gamma_{h\ell}} \Gamma_{h\ell} \frac{d\zeta_\ell^*}{dC_{ij}} \right).$$

We summarize the above procedure for computing the gradient for sorted SOFT top-$k$ operator in Algorithm 3. This naive implementation takes $\mathcal{O}(n^2 k)$ complexity, which is not efficient. Therefore, we modify the algorithm using the associative law of matrix multiplications, so that the complexity is lowered to $\mathcal{O}(nk)$. We summarize the modified algorithm in Algorithm 4.

We also include the `PyTorch` implementation of the forward pass and backward pass as shown below. The code is executed by creating an instance of `TopK_custom`, and the forward pass and the backward pass is run similar to any other `PyTorch` model.

---

**Algorithm 3** Gradient for Sorted Top-$k$

---

**Require:** $C \in \mathbb{R}^{n \times (k+1)}, \mu \in \mathbb{R}^n, \nu \in \mathbb{R}^{k+1}, \frac{d\mathcal{L}}{d\Gamma} \in \mathbb{R}^{n \times (k+1)}, \epsilon$

  Run forward pass to get $\Gamma$

  $\bar{\nu} = \nu[:-1], \bar{\Gamma} = \Gamma[:,:-1]$

  $\mathcal{K} \leftarrow \text{diag}(\bar{\nu}) - \bar{\Gamma}^T (\text{diag}(\mu))^{-1} \bar{\Gamma}$        # $\mathcal{K} \in \mathbb{R}^{k \times k}$

  $H1 \leftarrow (\text{diag}(\mu))^{-1} + (\text{diag}(\mu))^{-1} \bar{\Gamma} \mathcal{K}^{-1} \bar{\Gamma}^T (\text{diag}(\mu))^{-1}$        # $H1 \in \mathbb{R}^{n \times n}$

  $H2 \leftarrow -(\text{diag}(\mu))^{-1} \bar{\Gamma} \mathcal{K}^{-1}$        # $H2 \in \mathbb{R}^{n \times k}$

  $H3 \leftarrow (H2)^T$        # $H3 \in \mathbb{R}^{k \times n}$

  $H4 \leftarrow \mathcal{K}^{-1}$        # $H4 \in \mathbb{R}^{k \times k}$

  Pad $H2$ to be $[n, k+1]$ in the last column with value 0

  Pad $H4$ to be $[k, k+1]$ in the last column with value 0

  $[\frac{d\xi^*}{dC}]_{hij} \leftarrow [H1]_{hi} \Gamma_{ij} + [H2]_{hj} \Gamma_{ij}$        # $\frac{d\xi^*}{dC} \in \mathbb{R}^{n \times n \times (k+1)}$

  $[\frac{d\zeta^*}{dC}]_{\ell ij} \leftarrow [H3]_{\ell i} \Gamma_{ij} + [H4]_{\ell j} \Gamma_{ij}$        # $\frac{db^*}{dC} \in \mathbb{R}^{k \times n \times (k+1)}$

  Pad $\frac{d\zeta^*}{dC}$ to be $[k+1, n, k+1]$ with value 0

  $[\frac{d\mathcal{L}}{dC}]_{ij} \leftarrow \frac{1}{\epsilon}(-[\frac{d\mathcal{L}}{d\Gamma}]_{ij}\Gamma_{ij} + \sum_{h,\ell}[\frac{d\mathcal{L}}{d\Gamma}]_{h\ell}\Gamma_{h\ell}[\frac{d\xi^*}{dC}]_{hij} + \sum_{h,\ell}[\frac{d\mathcal{L}}{d\Gamma}]_{h\ell}\Gamma_{h\ell}[\frac{d\zeta^*}{dC}]_{\ell ij})$

---

---

**Algorithm 4** Gradient for Sorted Top-$k$, with reduced memory

---

**Require:** $C \in \mathbb{R}^{N \times (K+1)}, \mu \in \mathbb{R}^N, \nu \in \mathbb{R}^{K+1}, \frac{d\mathcal{L}}{d\Gamma} \in \mathbb{R}^{N \times (K+1)}, \epsilon$

  Run forward pass to get $\Gamma$

  $\bar{\nu} = \nu[:-1], \bar{\Gamma} = \Gamma[:,:-1]$

  $\mathcal{K} \leftarrow \text{diag}(\bar{\nu}) - \bar{\Gamma}^T (\text{diag}(\mu))^{-1} \bar{\Gamma}$        # $\mathcal{K} \in \mathbb{R}^{K \times K}$

  $\mu'_i = \mu_i^{-1}$

  $L \leftarrow (\text{diag}(\mu))^{-1} \bar{\Gamma} \mathcal{K}^{-1}$        # $L \in \mathbb{R}^{N \times K}$

  $G1 \leftarrow \frac{d\mathcal{L}}{d\Gamma} \odot \Gamma$        # $G1 \in \mathbb{R}^{N \times K}$

  $g1 \leftarrow [G1]\mathbf{1}_K, g2 \leftarrow [G1]^T\mathbf{1}_N$        # $g1 \in \mathbb{R}^N, g2 \in \mathbb{R}^K$

  $G21 \leftarrow (g1 \odot \mu').expand\_dims(1) \odot \Gamma$        # $G21 \in \mathbb{R}^{N \times (K+1)}$

  $G22 \leftarrow ((g1)^T L \bar{\Gamma}^T \odot \mu').expand\_dims(1) \odot \Gamma$        # $G22 \in \mathbb{R}^{N \times (K+1)}$

  $G23 \leftarrow -((g1)^T L).pad\_last\_entry(0).expand\_dims(0) \odot \Gamma$        # $G23 \in \mathbb{R}^{N \times (K+1)}$

  $G2 = G21 + G22 + G23$        # $G2 \in \mathbb{R}^{N \times (K+1)}$

  $g2 \leftarrow g2[:-1]$

  $G31 \leftarrow -(L(g2)).expand\_dims(1) \odot \Gamma$        # $G31 \in \mathbb{R}^{N \times (K+1)}$

  $G32 \leftarrow (\mathcal{K}^{-1}(g2)).pad\_last\_entry(0).expand\_dims(0) \odot \Gamma$        # $G32 \in \mathbb{R}^{N \times (K+1)}$

  $G3 = G31 + G32$        # $G3 \in \mathbb{R}^{N \times (K+1)}$

  $\frac{d\mathcal{L}}{dC} \leftarrow \frac{1}{\epsilon}(-G1 + G2 + G3)$

---

```python
def sinkhorn_forward(C, mu, nu, epsilon, max_iter):
    bs, n, k_ = C.size()

    v = torch.ones([bs, 1, k_])/(k_)
    G = torch.exp(-C/epsilon)
    if torch.cuda.is_available():
        v = v.cuda()

    for i in range(max_iter):
        u = mu/(G*v).sum(-1, keepdim=True)
        v = nu/(G*u).sum(-2, keepdim=True)

    Gamma = u*G*v
    return Gamma

def sinkhorn_forward_stablized(C, mu, nu, epsilon, max_iter):
    bs, n, k_ = C.size()
    k = k_-1

    f = torch.zeros([bs, n, 1])
    g = torch.zeros([bs, 1, k+1])
    if torch.cuda.is_available():
        f = f.cuda()
        g = g.cuda()

    epsilon_log_mu = epsilon*torch.log(mu)
    epsilon_log_nu = epsilon*torch.log(nu)
```

```
        def min_epsilon_row(Z, epsilon):
            return -epsilon*torch.logsumexp((-Z)/epsilon, -1, keepdim=True)

        def min_epsilon_col(Z, epsilon):
            return -epsilon*torch.logsumexp((-Z)/epsilon, -2, keepdim=True)

        for i in range(max_iter):
            f = min_epsilon_row(C-g, epsilon)+epsilon_log_mu
            g = min_epsilon_col(C-f, epsilon)+epsilon_log_nu

    Gamma = torch.exp((-C+f+g)/epsilon)
    return Gamma

def sinkhorn_backward(grad_output_Gamma, Gamma, mu, nu, epsilon):

    nu_ = nu[:,:,:-1]
    Gamma_ = Gamma[:,:,:-1]

    bs, n, k_ = Gamma.size()

    inv_mu = 1./(mu.view([1,-1]))    #[1, n]
    Kappa = torch.diag_embed(nu_.squeeze(-2)) \
            -torch.matmul(Gamma_.transpose(-1, -2) * inv_mu.unsqueeze(-2), Gamma_)    #[bs, k, k]

    inv_Kappa = torch.inverse(Kappa)    #[bs, k, k]

    Gamma_mu = inv_mu.unsqueeze(-1)*Gamma_
    L = Gamma_mu.matmul(inv_Kappa)    #[bs, n, k]
    G1 = grad_output_Gamma * Gamma    #[bs, n, k+1]

    g1 = G1.sum(-1)
    G21 = (g1*inv_mu).unsqueeze(-1)*Gamma    #[bs, n, k+1]
    g1_L = g1.unsqueeze(-2).matmul(L)    #[bs, 1, k]
    G22 = g1_L.matmul(Gamma_mu.transpose(-1,-2)).transpose(-1,-2)*Gamma    #[bs, n, k+1]
    G23 = - F.pad(g1_L, pad=(0, 1), mode='constant', value=0)*Gamma    #[bs, n, k+1]
    G2 = G21 + G22 + G23    #[bs, n, k+1]

    del g1, G21, G22, G23, Gamma_mu

    g2 = G1.sum(-2).unsqueeze(-1)    #[bs, k+1, 1]
    g2 = g2[:,:-1,:]    #[bs, k, 1]
    G31 = - L.matmul(g2)*Gamma    #[bs, n, k+1]
    G32 = F.pad(inv_Kappa.matmul(g2).transpose(-1,-2), pad=(0, 1), mode='constant', value=0)*Gamma
#[bs, n, k+1]
    G3 = G31 + G32    #[bs, n, k+1]

    grad_C = (-G1+G2+G3)/epsilon    #[bs, n, k+1]
    return grad_C

class TopKFunc(Function):
    @staticmethod
    def forward(ctx, C, mu, nu, epsilon, max_iter):

        with torch.no_grad():
            if epsilon>1e-2:
                Gamma = sinkhorn_forward(C, mu, nu, epsilon, max_iter)
                if bool(torch.any(Gamma!=Gamma)):
                    print('Nan appeared in Gamma, re-computing...')
                    Gamma = sinkhorn_forward_stablized(C, mu, nu, epsilon, max_iter)
            else:
                Gamma = sinkhorn_forward_stablized(C, mu, nu, epsilon, max_iter)
            ctx.save_for_backward(mu, nu, Gamma)
            ctx.epsilon = epsilon
        return Gamma

    @staticmethod
    def backward(ctx, grad_output_Gamma):

        epsilon = ctx.epsilon
        mu, nu, Gamma = ctx.saved_tensors
        # mu [1, n, 1]
        # nu [1, 1, k+1]
        #Gamma [bs, n, k+1]
        with torch.no_grad():
            grad_C = sinkhorn_backward(grad_output_Gamma, Gamma, mu, nu, epsilon)
        return grad_C, None, None, None, None

class TopK_custom(torch.nn.Module):
    def __init__(self, k, epsilon=0.1, max_iter = 200):
        super(TopK_custom1, self).__init__()
        self.k = k
        self.epsilon = epsilon
        self.anchors = torch.FloatTensor([k-i for i in range(k+1)]).view([1,1, k+1])
        self.max_iter = max_iter

        if torch.cuda.is_available():
            self.anchors = self.anchors.cuda()

    def forward(self, scores):
        bs, n = scores.size()
```

```
scores = scores.view([bs, n, 1])

#find the -inf value and replace it with the minimum value except -inf
scores_ = scores.clone().detach()
max_scores = torch.max(scores_).detach()
scores_[scores_==float('-inf')] = float('inf')
min_scores = torch.min(scores_).detach()
filled_value = min_scores - (max_scores-min_scores)
mask = scores==float('-inf')
scores = scores.masked_fill(mask, filled_value)

C = (scores-self.anchors)**2
C = C / (C.max().detach())

mu = torch.ones([1, n, 1], requires_grad=False)/n
nu = [1./n for _ in range(self.k)]
nu.append((n-self.k)/n)
nu = torch.FloatTensor(nu).view([1, 1, self.k+1])

if torch.cuda.is_available():
    mu = mu.cuda()
    nu = nu.cuda()

Gamma = TopKFunc.apply(C, mu, nu, self.epsilon, self.max_iter)

A = Gamma[:,:,:self.k]*n

return A, None
```

# C   Experiment Settings

## C.1   $k$NN

The settings of the neural networks, the training procedure, and the number of neighbors $k$, and the tuning procedures are similar to Grover et al. (2019). The tuning o $\epsilon$ ranging from $10^{-6}$ to $10^{-2}$. Other settings are shown in Table 3.

**Table 3:**  Parameter settings for $k$NN experiments.

| Dataset | MNIST | CIFAR-10 |
|---|---|---|
| $k$ | 9 | 9 |
| $\epsilon$ | $10^{-3}$ | $10^{-5}$ |
| Batch size of query samples | 100 | 100 |
| Batch size of template samples | 100 | 100 |
| Optimizer | SGD | SGD |
| Learning rate | $10^{-3}$ | $10^{-3}$ |
| Momentum | 0.9 | 0.9 |
| Weight decay | $5 \times 10^{-4}$ | $5 \times 10^{-4}$ |
| Model | 2-layer convolutional network | ResNet18 |

Note that $f_\theta$ is a feature extraction neural network, so that model specified in the last row of Table 3 does not contain the final activation layer and the linear layer.

**Baselines.** In the baselines, the results of $k$NN, $k$NN+PCA, $k$NN+AE, $k$NN+NeuralSort is copied from Grover et al. (2019). The result of RelaxSubSample is copied from Xie and Ermon (2019).

The implementation of $k$NN+Cuturi et al. (2019) is based on Grover et al. (2019). Specifically, the outputs of the models in Cuturi et al. (2019) and Grover et al. (2019) are both doubly stochastic matrices. So in the implementation of $k$NN+Cuturi et al. (2019), we adopt the algorithm in Grover et al. (2019), except that we replace the module of computing the doubly stochastic matrix to be the one in Cuturi et al. (2019). We extensively tuned $k$, $\epsilon$ and the learning rate, but cannot achieve a better score for this experiment.

The baselines $k$NN+Softmax $k$ times, $k$NN+pretrained CNN, and CE+CNN adopts the identical neural networks as our model. We remark that the scores reported in Grover et al. (2019) for CNN+CE are 99.4% for MNIST and 95.1% for CIFAR-10. However, our experiments *using their code* cannot reproduce the reported scores: and the scores are 99.0% and 90.9%, respectively. Therefore, the reported score for MNIST is implemented by us, and the score for CIFAR-10 is copied from He et al. (2016).

## C.2 Beam Search

**Algorithm.** We now elaborate how to backtrack the predecessors $E^{(1:t),r}$ for an embedding $E^{(t+1),\ell}$, and how to compute the likelihood $\mathcal{L}_{\text{s}}(E^{(1:t+1),\ell})$, which we have omitted in Algorithm 2. Specifically, in standard beam search algorithm, each selected token $\widetilde{y}^{(t+1),\ell}$ is generated from a specific predecessor, and thus the backtracking is straightforward. In beam search with sorted SOFT top-$k$ operator, however, each computed embedding $E^{(1:t),r}$ is a weighted sum of the output from all predecessors, so that it is not corresponding to one specific predecessor. To address this difficulty, we select the predecessor for $E^{(t+1),\ell}$ with the largest weight, i.e.,

$$(o, r) = \underset{(j,i)}{\operatorname{argmax}} \, A^{(t),\epsilon}_{ji,\ell}.$$

This is a good approximation because $A^{(t),\epsilon}$ is a smoothed 0-1 tensor, i.e., for each $\ell$, there is only one entry that is approximately 1 in $A^{(t),\epsilon}_{:,:,\ell}$, while the others are approximately 0. The likelihood is then computed as follows

$$\mathcal{L}_{\text{s}}(E^{(1:t+1),\ell}) = \mathcal{L}_{\text{s}}(E^{(1:t),r})\mathbb{P}(y^{t+1} = \omega_o|\widetilde{h}^{(t),r}(E^{(1:t),r})).$$

**Implementation.** The implemented model is identical to Bahdanau et al. (2014). Different from Bahdanau et al. (2014), here we also preprocess the data with *byte pair encoding* (Sennrich et al., 2015).

We adopt beam size 5, teacher forcing ratio $\rho = 0.8$, and $\epsilon = 10^{-1}$. The training procedure is as follows: We first pretrain the model with teacher forcing training procedure. The pretraining procedure has initial learning rate 1, learning rate decay 0.1 starting from iteration $5 \times 10^5$ for every $10^5$ iterations. We pretrain it for $10^6$ iterations in total. We then train the model using the combined training procedure for $10^5$ iterations with learning rate 0.05.

## C.3 Top-$k$ Attention for Machine Translation

We apply SOFT top-$k$ operator to yield sparse attention scores. Attention module is an integral part of various natural language processing tasks, allowing modeling of long-term and local dependencies. Specifically, given the vector representations of a source sequence $s = [s_1, \cdots, s_N]^\top$ and target sequence $y = [y_1, \cdots, y_M]^\top$, we compute the alignment score between $s_i$ and $y_j$ by a compatibility function $f(s_i, y_j)$, e.g., $f(s_i, y_j) = s_i^\top y_j$, which measures the dependency between $s_i$ and $y_j$. A softmax function then transforms the scores $[f(s_i, y_j)]_{i=1}^N$ to a sum-to-one weight vector $w_j$ for each $y_j$. The output $o_j$ of this attention module is a weighted sum of $s_i$'s, i.e., $o_j = w_j^\top s$.

The attention module described above is called the soft attention, i.e., the attention scores $w_j$ of $y_j$ is not sparse. This may lead to redundancy of the attention (Zhu et al., 2018; Schlemper et al., 2019). Empirical results show that hard attention, i.e., enforcing sparsity structures in the score $w_j$'s, yields more appealing performance (Shankar et al., 2018). Therefore, we propose to replace the softmax operation on $[f(s_i, y_j)]_{i=1}^N$ by the standard top-$k$ operator to select the top-$k$ elements. In order for an end-to-end training, we further deploy SOFT top-$k$ operator to substitute the standard top-$k$ operator. Given $[f(s_i, y_j)]_{i=1}^N$, the output of SOFT top-$k$ operator is denoted by $A_j^\epsilon$, and the weight vector $w_j$ is now computed as

$$w_j = \operatorname{softmax}([f(s_1, y_j), \ldots, f(s_N, y_j)]^\top + \log A_j^\epsilon).$$

Here $\log$ is the entrywise logarithm. The output $o_j$ of the attention module is computed the same $o_j = w_j^\top s$. Such a SOFT top-$k$ attention will promote the top-$k$ elements in $[f(s_i, y_j)]_{i=1}^N$ to be even larger than the non-top-$k$ elements, and eventually promote the attention of $y_j$ to focus on $k$ tokens in $s$.

### C.3.1 Experiment

We evaluate the proposed top-$k$ attention on WMT2016 English-German dataset. Our implementation and settings are based on Klein et al. (2017)[5]. For a fair comparison, we implement a standard soft attention using the same settings as the baseline. The details are provided in Appendix C.

(a) $\mathcal{I} = \{0, 1, 2\}$.  (b) $\mathcal{I} = \{2, 3, 4\}$.

**Figure 7:** Illustration of the gradient of the SOFT top-$k$ operators. The arrows represent the direction and magnitude of the gradient. The orange dots corresponds to the ground truth elements.

**Results.** As shown in Table 4, the proposed SOFT top-$k$ attention training procedure achieves an improvement in BLEU score of approximately $0.8$. We visualize the top-$k$ attention in Figure 6. The attention matrix is sparse, and has a clear semantic meaning – "truck" corresponds to "Lastwagen", "blue" corresponds to "blauen", "standing" corresponds to "stehen", etc.

**Table 4:** BLEU scores on WMT'16.

| Algorithm | BLEU |
|---|---|
| Proposed Top-$k$ Attention | **37.30** |
| Soft Attention | 36.54 |

## D  Visualization of the Gradients

In this section we visualize the computed gradient using a toy example mimicking the settings of $k$NN classification. Specifically, we input 10 scores computed from 10 images, i.e., $\mathcal{X} = \{0, 1, 2, \cdots, 9\}$, into the SOFT top-$k$ operator, and select the top-3 elements. Denote the indices of the images with the same labels as the query sample as $\mathcal{I}$. Similar to $k$NN classification, we want to maximize $\sum_{i \in \mathcal{I}} A_i^\epsilon$.

**Figure 6:** Visualization of the top-$K$ attention.

We visualize the gradient on $\mathcal{X}$ with respect to this objective function in Figure 7. In Figure 7(a), $\mathcal{I}$ is the same as the indices of top-3 scores. In this case, the gradient will push the gap between the top-3 scores and the rest scores even further. In Figure 7(b), $\mathcal{I}$ is different from the indices of top-3 scores. In this case, the scores corresponding to $\mathcal{I}$ are pushed to be smaller, while the others are pushed to be larger.