[Reviews · NeurIPS 2020]

Review 1

Summary and Contributions: The authors aim at providing an (approximate) differentiable arg-top-k algorithm. For this they rephrase the problem into an entropy regularized optimal transport problem, which can be solved by the (differentiable) Sinkhorn algorithm. Edit after rebuttal: I have read the authors' response and the other reviews. My main questions have been addressed. I have updated my score.

Strengths: The paper is well written, visualized and is easy to understand. Also important experiments were conducted and compared to other baselines.

Weaknesses: The formulation into optimal transport seems a bit arbitrary as very simple choices for all occuring parameters were used, e.g. the loss function and representation of the categoris , e.g. 0,1 on the line. If such simple heuristics work, why not directly using an efficient classical (near linear time and space) sorting algorithm for real numbers, breaking up the permutation into transpositions and using a relaxed version for the gradients, e.g. something like s'(x2-x1), where s is any sigmoidal function, cf. figure 1. This could work equally well or even more efficient. Please do such comparisons and comment! It was not discussed how these choices effect the algorithm. Also why other differentiable sorting algorithms perform worse is not discussed. Please do so!

Correctness: The methods seem correct.

Clarity: The paper is clearly written. The notations could be improved.

Relation to Prior Work: Other methods seem to be well cited and compared to. A bit more discussion why the other methods perfom worse would be helpful though.

Reproducibility: Yes

Additional Feedback:


Review 2

Summary and Contributions: This paper proposes Soft-TopK as a differentiable version of TopK such that it can be integrated in many end-to-end training pipelines that rely on the TopK operations. The idea is to consider Top-K as an optimal transport problem between input sequence and output indicators, and Entropic OT can be used to get a bias but smooth solution. The experiments are performed on 3 tasks: knn image classifier, beam search MT, and sparse attention based MT.

Strengths: Using OT to solve top k problem is an interesting idea. Even though similar ideas have been proposed for ranking and sorting, it seems the extension to top k operation is new. The bias control (Theorem 2) and efficient backward implementation look sound.

Weaknesses: My main concern is in evaluation. Please see my detailed comments below.

Correctness: Mostly correct. But I don’t have enough time/bandwidth to check all the proofs and the code carefully.

Clarity: yes

Relation to Prior Work: yes

Reproducibility: Yes

Additional Feedback: Evaluation: The paper made a lot of effort on comparing the proposed soft-topk with many competing top k approximation algorithms in the task of KNN image classification. However, the evaluations on the other two tasks are questionable. For example, on the beam search MT task, the paper compared only with previous maximum likelihood based MT or other training methodologies. Since the paper mainly proposes a new top-k approximator, it should compare with other top-k approximators on the same MT framework instead of comparing MT with different loss functions. Similarly, for top-k attention based MT, there is no comparison with other competing top-k approximation methods. Also, a simple hard top-k attention can be applied in this task. It’s interesting to compare with it too. ------------------- Post-rebuttal: I have read the comments from all the reviewers and the authors' rebuttal. It's an incremental work, but the algorithm and the results will be useful for the community. I think my original score represents the general view of this paper and therefore I keep my rating the same.


Review 3

Summary and Contributions: The paper presents a smoothing of the top-k operator by adding an entropic regularized to the optimization problem defining the top-k operator. The use of an entropic regularized allows the authors to obtain a differentiable surrogate of the mapping. Several experiments illustrate the proposed algorithm on different tasks

Strengths: The surrogate presented by the authors is differential in comparison to other works that only get continuous but potentially non-smooth surrogates of the argmax. Moreover, they present an interesting bound between the result of the top-k operator and their algorithm. I believe this latter result can be interested for further research on the smoothing on argmax operators.

Weaknesses: - The algorithm would not scale to structured prediction tasks where the space in which the argmax is computed is of exponential size. - The overall beam search with soft top-k method appears to be a sequence of tricks rather than a principled method. It may be related to Differentiable Dynamic Programming for Structured Prediction and Attention, Mensch, Blondel, 2018 - The authors quickly discard the possibility of ties in the top-k operator. Yet I do not see any reasons why it cannot happen in theory.

Correctness: The main theorems are correct. The experimental results are missing standard errors. Even if the other works do not provide standard errors, the present work may provide some.

Clarity: The authors let multiple results to be done by the readers. While these results are indeed feasible, the paper would benefit from clarifications to make it accessible to a large audience. Overall the paper is not well written to be easily readable. Examples: l 132 "One can check that" Just do it or point to a reference l. 172 y_i \in \{0,1\}^M. l. 180 There is a mistake in the definition of the loss there should not be a transpose for Y_{\j} l 448 "It is straightforward" just explain in a few lines. l 556 Define xi and eta before using it l 461: convexity -> concavity l 465: define y_i l 469: why is there an additional delta C_j? l 470: "easy to verify that", just do it or point to a reference l 518: Z -> Z/n l 530: Gamma(I_j) - Gamma(I*) -> B_j Overall proof read: sometimes K is used instead fo k and N is used instead of n (e.g. in the proof lemma 3

Relation to Prior Work: The idea of the paper is exactly the same as the one of Cuturi et al 2019, except that it focuses on top-k operators. The implementation takes advantage of algebraic tools to compute gradients by the implicit function theorem and a theorem about the approximation of the soft top-k operator is given. A paragraph on beam search is added but not clearly justified. Moreover it is slightly unclear which method from Cuturi et al is chosen. In particular is it the quantile loss or the top-l loss?

Reproducibility: Yes

Additional Feedback: Some comments: l. 117 The entropic OT is surely not more computationally friendly than a a top-k operator that simply sorts the vector. l. 180 Can you write down explicitly the loss that is used with the top-k and explain its relevance statistically ? Same for the beam-search method, the present work seems to be a sequence of ad-hoc definitions rather than a principled objective. In particular it is important to make the optimization objective clear to enable future comparisons. Can the authors clearly distinguish their contributions from the ones of Cuturi et al, 2019? It seems that the implementation of the authors is potentially faster, which should be highlighted. Other than that the applications may be original if they were clearly motivated as asked above. ==== After rebuttal ==== I thank the authors for their answers to the questions raised above. I have read the author's rebutttal. 1. This is an incremental paper. The idea was thoroughly presented by (Cuturi et al, 2019) for a variety of tasks. The key contributions in comparison to (Cuturi et al, 2019) should be clearly highlighted for the sake of future references. The present rebuttal answered partially this point. In particular it is unclear to me if the present top-k operator is better or worse than a quantile based approach. 2. The proof of the approximation result is not particularly original. Yet it is correct and the result may be of potential interest. 3. The applications are hard to grasp. In particular the beam search application is not well-motivated by an optimization objective. However, the current experimental results demonstrate clearly the benefits of the approach. For the final version, if space is allowed the authors should consider detailing the overall approach (at least in the supplementary) by precising the objectives. 4. The main argument for me to increase my score is that their code can easily be used. It is implemented as a Pytorch differentiable function with a specific implementation of the backward pass. I tested it and I was able to compare their differentiation against a differentiation through the iterations of Sinkhorn. The algebra tricks used to derive the gradient is another interesting increment over (Cuturi et al, 2019). Overall their code could be used easily by other practitioners and would be an interesting new layer in the library of Pytorch. Overall this is an incremental paper. Yet, some of the contributions may be interesting or the community. I have increased my score accordingly.


Review 4

Summary and Contributions: * The authors propose a differentiable surrogate to the top-k operator by leveraging ideas from optimal transport. Significance: High * The authors show how this can be used for end-to-end training of k-NN classification and a differentiable variant of beam search. Significance: Medium

Strengths: The problem is well-motivated and highly relevant. The claims made by the paper are sound (although I have concerns about the beam search, see W2). The paper presents an useful application of optimal transport in differentiable programming. While this has a precedent, the applications are useful enough to warrant interest. Further, empirical comparisons are made to a number of strong baselines.

Weaknesses: W1. Some sensitivity analyses are not performed. a) How many Sinkhorn iterations do you use? What is the sensitivity of the proposed method to varying this parameter? This is important, since the backpropagation assumes that a highly accurate solution to the entropic OT problem has been found, while Sinkhorn iterations only return an approximate solution. b) The computation of k-NN loss from a batch of template examples (as opposed to the entire dataset) introduces a bias, and the stochastic gradients obtained are no longer unbiased estimates of the true gradients. In this case, a sensitivity study must be performed against the size of the template batch. c) It would be interesting to see the sensitivity of the proposed method to the teacher forcing ratio and the smoothing parameter epsilon. W2. The motivation behind some of the design choices for the differentiable beam search requires more discussion and justification. Some issues I see are: a) The prefix r in eq. (9) involves computing the argmax for the prefix (while the next token is computed as a weighted sum in a differentiable manner). The computation graph created thus involves the argmax, which is not differentiable. Why do the authors not take the approach of computing the embedding E^{1:t} as a weighted sum as well? b) Hidden states are computed as a weighted sum. However, the hidden state dynamics and the next token are not related linearly. Therefore, the hidden state which gives rise to the average token considered in eq. (9) might not correspond to the one considered in eq. (10). c) The loss L_{soft} in line 262 is a token-level loss. It is not clear to me why matching a token given the decoded prefix is the right loss. Is it not more natural to use a sequence level loss? (e.g., Edunov et. al. arXiv:1711.04956). d) What is the strategy used for length mismatch between decoded sequences and the gold sequence? Especially, since the loss L_{soft} is defined token-wise.

Correctness: Yes

Clarity: The paper is reasonably well-written and easy to read.

Relation to Prior Work: Yes. However, the following would be good to have: * A detailed discussion of how the proposed work connects to ranking would be helpful, especially for the case when K=O(n). The current discussion involves only the computational aspects. * How does the proposed approach with K=1 compare to existing approaches for backpropagation through argmax/argmin of a vector? Further, if the proposed approximation be computed in closed-form in this simple case, it might be illuminating to discuss it.

Reproducibility: Yes

Additional Feedback: Post-author-feedback ——————————- Thanks to the authors for their detailed response and the sensitivity analyses. Many of my questions have been answered. I would request the authors to add these details to the paper. I have updated my score accordingly. W2a: It is not obvious to me how this is possible. It would help to add an explanation about this (perhaps in the supplement). W2b: It would help to note the approximation used in the paper My concerns regarding related work are not addressed in the rebuttal. I trust the authors to address these in the final version. ======================================= * Some details are unclear: - Is the same batch used for both the template samples and the template samples? - Which optimizers are used? - What is the running time of the experiments and what is the hardware used? * Regarding the proof of Theorem 1: - L is not strongly convex but only strictly convex (actually, concave) in lines 461-462 - The invocation of implicit function theorem requires checking that its assumptions are true. For instance, see Theorem 2 of the paper Luise et. al. (2018) referred to by the authors. Minor Comments: * Line 178: \omega should be \theta? * Line Line 220: should it be P(y^{t+1} = z_i | ...)? * The use of dx/dt versus \partial x /\partial t is not uniform throughout the paper. In particular, see page 19. * It might be worth clarifying that dx/dt in the multivariate case refers to the Jacobian, and not its transpose. * page 19: a, b are used in place of \xi, \zeta in multiple places.

[Author Response · NeurIPS 2020]

**To Reviewer #1**: Choosing any two distinct scalars on the line is equivalent. Specifically, the minimizer in Proposition 1 will not change if another two distinct scalers are chosen. Therefore, we choose $0, 1$ to simplify the discussion.

We experimented on using Sigmoid functions, and it does not work. We approximate the indicator vector $A^\epsilon$ by $s(x_i - x_{\sigma_k})$, where $i = 1, \cdots, n$, and $x_{\sigma_k}$ is the $k$-th smallest input. We first take $s(\cdot)$ as a standard Sigmoid function, and it quickly runs into numerical issues even with extensive parameter tuning. We then try a hard Sigmoid function (arXiv:1811.03378) with different slopes. The best performance is 27% on CIFAR10 (worse than a simple $k$NN). The gradient computed cannot provide effective guidance for the parameter updates.

**To Reviewer #2**: For beam search, $n$ is very large. NeuralSort and Cuturi (2019) require $O(n^2)$ memory, which is not affordable. We tried Softmax for $k$ times (which is also proposed by us). The performance is comparable to SOFT, but it often runs into numerical issues when computing the gradient since it nests Softmax for many times. The BLEU score for a hard Top-k attention is 37.02 (SOFT 37.30). We will add more discussions in the next version.

**To Reviewer #3**: Our algorithm scales linearly w.r.t. the input size, which is not larger than any other algorithms. Furthermore, we already applied our method to a structured prediction task, i.e., machine translation, where we use beam search to search over all combinations of the tokens.

The proposed beam search method is a principled way to close the exposure bias between the training and the inference, originating from curriculum learning (Bengio, 2015, arXiv:1506.03099). Traditionally, in training, the input to the decoder is the gold sequence, while in the inference, its input is sequences decoded by beam search. So we incorporate beam search into training to close this gap.

Furthermore, the proposed method achieves a significant improvement over a very strong baseline (Bahdanau, 2014) with nearly identical hyperparameters, suggesting the proposed method is more than tricks. (Note that BLEU of our implemented Bahdanau is 35.38, which is as far as we know the best score for RNN-based seq2seq single models. The original Bahdanau is only 28.45).

We did not consider ties because this rarely happens in practice – the input of SOFT should be float number with at least $10^{-5}$ precision. Furthermore, if there are ties, it does not matter which to choose among the ties for machine learning algorithms. We also remark that our method can apply to cases with ties naturally: Instead of returning a smoothed indicator vector, the entries for two tied scalars will be approximately 0.5.

We highlight our key contributions over Cuturi, (2019):
1. SOFT has $O(n)$ complexity while the sorting algorithm in (Cuturi, 2019) is $O(n^2)$ (line 298).
2. We derive a fast and memory-efficient way to compute the gradient of the top-$k$ operation (line 319).
3. We prove that the approximation error can be properly controlled (Theorem 2).
4. We propose novel applications, i.e., image classification with kNN and beam search training scheme (Section 4, Section 5). We did not adopt the quantile loss or the top-$l$ loss. We propose new losses.

**To Reviewer #5**: We will include more discussions on the sensitivity and motivation in the next version.

W1 a) For beam search, we use 100 inner iterations for a relatively large $\epsilon$ (0.05). We use 2000 inner iterations for $k$NN, since we adopt a very small $\epsilon$ ($10^{-5}$ for CIFAR10). For very small $\epsilon$ ($10^{-5}$), we do observe significant performance drop with fewer Sinkhorn iterations. For larger $\epsilon$, the performance is not sensitive to the number of iterations.

b) The sensitivity result of CIFAR 10 for template sample size is as follows:

| Template batch size | 100 (current) | 200 | 300 | 400 | 500 |
|---|---|---|---|---|---|
| Acc. change | 0 | $+0.40$ | $+0.34$ | $+0.43$ | $-0.18$ |

The bias introduced by small template batch size does not significantly affect the final performance.

c) Current teacher forcing ratio $\rho$ is 0.8. With $\rho = 0.7$, the BLUE score will decrease 0.25. Currently we have $\epsilon = 0.05$. With $\epsilon = 0.1$, BLUE score will decrease 0.13.

W2 a). We compute the maximum likelihood of the predicted sequence in $\mathcal{L}_{\text{SOFT}}$ (Equation (9)). This can be realized by a max operation, which is differentiable. Note that we don't need to use argmax operation to find the index $r$. Accordingly, there is no need to compute a weighted linear combination of the embedding.

b). The exact correspondence between the hidden state and the token can be very complicated. An explicit characterization requires sophisticated tools. On the contrary, we approximate the correspondence using a simple linear function. The experimental results indicate that the performance (Table 2) is already superior over existing methods.

c). A major benefit of the proposed decoding step is that the un-picked tokens are not abandoned in the computational graph. Instead, every decoded token is involved in the later computation. Therefore, by comparing the weighted sum of embedding to the gold token, we encourage the weight corresponding to the gold token to be larger, which is connected to all former decoded tokens in the computational graph. As a result, although the proposed loss appears to be token-level, we are essentially optimizing over all possible combinations of tokens. It is not necessary to adopt more complicated losses.

d) We pad the shorter sentence (the decoded and the gold sequence) with <EOS> before feeding them into NLL.

[Meta-Review · NeurIPS 2020]

The reviews for this paper were positive overall. The paper presents a smoothing technique for the top-k operator via the addition of an entropic regularization. The proposed smoothed counterpart departs from previous proposals in that it may be continuous yet non-smooth. arg-max. The contributions were considered 'well-motivated' and 'highly relevant'. The ease of use of the PyTorch implementation uploaded along with the paper was particularly appreciated. We recommend to take the reviewers' comments and suggestions into account while preparing the camera ready final version of the paper. The authors may pay particular attention to delineate clearly the contributions of the paper from previous papers on the subject (as well as the originality of their contributions). Accept.